# Complexions at the iron-magnetite interface

Xuyang Zhou [1] ✉, Baptiste Bienvenu [1] ✉, Yuxiang Wu[1], Alisson Kwiatkowski da Silva [1], Colin Ophus [2] & Dierk Raabe[1] ✉

Synthesizing distinct phases and controlling crystalline defects are key concepts in materials design. These approaches are often decoupled, with the former grounded in equilibrium thermodynamics and the latter in nonequilibrium kinetics. By unifying them through defect phase diagrams, we can apply phase equilibrium models to thermodynamically evaluate defects—including dislocations, grain boundaries, and phase boundaries—establishing a theoretical framework linking material imperfections to properties. Using scanning transmission electron microscopy (STEM) with differential phase contrast (DPC) imaging, we achieve the simultaneous imaging of heavy Fe and light O atoms, precisely mapping the atomic structure and chemical composition at the iron-magnetite ($Fe/Fe_3O_4$) interface. We identify a well-ordered two-layer interface-stabilized phase state (referred to as complexion) at the $Fe[001]/Fe_3O_4[001]$ interface. Using density-functional theory (DFT), we explain the observed complexion and map out various interface-stabilized phases as a function of the O chemical potential. The formation of complexions increases interface adhesion by 20% and alters charge transfer between adjacent materials, impacting transport properties. Our findings highlight the potential of tunable defect-stabilized phase states as a degree of freedom in materials design, enabling optimized corrosion protection, catalysis, and redox-driven phase transitions, with applications in materials sustainability, efficient energy conversion, and green steel production.

Iron oxides, ubiquitous in nature, science and engineering as well as integral to Earth's geological evolution, have far-reaching impacts on human civilization[1-3]. Their high importance includes topics such as heterogeneous catalysis[4,5], corrosion[6], spintronics[7], magnetic recording[8], energy[9], medicine[10], extraplanetary and terrestrial geophysics[2], geology[11,12], mining[13] and the emerging field of sustainable metallurgy[14-17]. The interfaces that connect iron oxides with adjacent phases play crucial roles for thermodynamics, kinetics, properties and performance across many types of practical applications[5,6,15,16,18-27]. Gaining deep understanding of the phenomena taking place at and across these interfaces is essential for fully exploiting their potential and controlling the thermodynamics and kinetics involved, leveraging the desired structure and chemistry for optimal performance, or respectively, better understanding in each of these specific fields.

Deciphering the unexpected complexity of their atomistic and chemical structure makes such interfaces accessible to manipulation, for example with respect to their role and properties in chemical redox processes. The iron oxide surface, for example, undergoes reconstruction—a process predominantly governed by Tasker's criteria for surface polarity[28] and by the prevalence of oxygen vacancies[19]. The atomic configurations of the surface terminations are not fixed, but they adapt based on the chemical potential of O in their immediate environment[19]. The interfaces between oxides and other phases are not merely kinematic entities with predetermined geometrical and chemical conditions. Instead, they can be subject to complex chemical, structural and electronic variations and reconstructions beyond singular atomistically sharp transition features, as revealed by recent observations[19,29].

[1]Max-Planck-Institut for Sustainable Materials (Max-Planck-Institut für Eisenforschung), Max-Planck-Straße 1, Düsseldorf, Germany. [2]National Center for Electron Microscopy, The Molecular Foundry, Lawrence Berkeley National Laboratory, Berkeley, USA. ✉e-mail: x.zhou@mpie.de; b.bienvenu@mpie.de; raabe@mpie.de

One particularly interesting class of interface reconstructions is the so-called complexion[30,31]. These have been described as thermodynamic entities[32] that form a compliant "mediating" layer with generic structural and chemical phase-like features between adjacent phases, different from the kinematically expected interface structure and chemistry. These complexions were suggested to not just represent a kinetic phenomenon but a thermodynamic one, that is only existent in the presence of adjacent phases, but not in conventional bulk phase diagrams, which are, per definition, defect-free representations of equilibrium phase states as a function of concentration, temperature, chemical potential and/or pressure. Although the "complexion" terminology has gained acceptance in various applications[29,33–36], its extend still invites debate[37,38]. In this work, we consistently use "complexion" aligning with our experimental observations and recent reviews[30,31] to describe these interfacial phenomena accurately. The atomic structure and chemistry at these interfaces differ significantly from their bulk counterparts and can even undergo structural and chemical transitions, similar to bulk phase transformations[29,33,35,36,39–42]. Given that the chemical potentials of all species $i$ involved, denoted as $\mu_i$, remain constant across complexions adjacent to phases, it becomes feasible to construct complexion diagrams using $\mu_i$ as the primary variable, alongside appropriate thermodynamic parameters that govern the stability and transitions of such defect structures[18,26,43–48]. Such a complexion diagram becomes an invaluable tool for investigating defect phases and the transitions among them, bringing the fields of lattice defects and thermodynamics closely together.

The importance of structural and chemical features of these interfacial reconstructions, or defect phase states, has been shown for a number of systems. The Dillon-Harmer complexions, categorized into six distinct types and originally observed in undoped and doped $Al_2O_3$ ceramics, helped to elucidate the mechanisms underlying the longstanding question of abnormal grain growth in inorganic systems[30]. Rare-earth elements, such as La, preferentially segregate to the grain boundaries of $Si_3N_4$ to form nanometer-scale amorphous inter-granular films, essential in developing a microstructure with elongated grains that play a key role in tuning mechanical properties[49,50]. In addition to ceramics, metallic systems also exhibit comparable phenomena, such as the formation of a bi-layer interfacial phase of Bi at Ni grain boundaries, a characteristic associated with the susceptibility to liquid metal embrittlement[41]. Nanometer-thick amorphous intergranular films have also been shown to reach equilibrium at the $Au/Al_2O_3$ interface[29]. These examples illustrate the broad significance of complexions in materials science and beyond, potentially leveraging substantial impact on material properties under various types of boundary conditions.

While studying the complexities of interface structures yield significant insights, their detailed characterization–specifically for iron oxides–poses considerable challenges, potentially leading to their under-representation in earlier investigations. Conventional scanning transmission electron microscopy (STEM) methods such as high-angle annular dark field (HAADF) imaging have limitations in resolving light elements, like O, against heavier ones, like Fe. In addressing this challenge, a recent breakthrough with the differential phase contrast (DPC) - four-dimensional STEM (4DSTEM) method has emerged, significantly enhancing the atomic-scale mapping of both structural and chemical features in such reconstructed regions, where imaging of these details is required to reveal the thermodynamic nature of interfacial phase states[51–56]. This technique advances the study of complex interfaces at high spatial resolution, offering insights into their structures and properties[55]. The DPC-4DSTEM approach enables the reconstruction of charge-density maps from the collected data, providing phase contrast that correlates linearly with the mass of the elements. This enables direct spatial resolution of light atoms together with heavier atoms[57,58], such as O within oxides and at interfaces. In the following sections, we present how DPC-4DSTEM probing advances

our understanding of complex phenomena taking place at the iron/iron oxide interfaces, offering insights that can lead to informed strategies for material design and a better understanding of geophysical phenomena, catalysis and sustainable metallurgical synthesis.

## Results and discussion

### Mapping the structure of Fe[001]/Fe₃O₄[001] interface at the atomic scale

We conducted DPC-4DSTEM measurements to study the atomic structure of the Fe/iron oxide interface, formed by the epitaxial growth of multi-layer thin films on single-crystal MgO substrates at 300 °C. We chose the thin film method to fabricate interfaces with a specific orientation relationship, aiming to create an ideal sample condition for high-resolution imaging. Further details on sample preparation via physical vapor deposition (PVD) can be found in "Methods" Section Synthesis and Supplementary Figs. 1–3.

Using the magnetite ($Fe_3O_4$) oriented in the [110] direction, we demonstrated the strength of the DPC-4DSTEM method in resolving both heavy Fe and light O atomic columns simultaneously. Figure 1ai–iv showcase the corresponding experimentally reconstructed virtual dark-field image, electric field vector map, projected electrostatic potential map, and charge-density map from the DPC-4DSTEM data set. The virtual dark-field image (see Fig. 1ai) reveals the position of the Fe atomic columns, but it lacks sufficient resolution to distinguish lighter O atomic columns. On the other hand, the projected potential map (see Fig. 1aiii) and the charge-density map (see Fig. 1ai–v) can clearly resolve both types of atomic columns. In these three images, the positions of parts of the O atomic columns are indicated by white arrows. Additional reconstructed images, including the center of mass, are presented in "Methods" Section Characterization and Supplementary Figs. 4–6. Since the charge-density map provides the clearest simultaneous representation of both the heavy Fe and the lighter O atoms, we will primarily use it to investigate the atomic structure of the Fe/iron oxide interface in the subsequent discussion.

Figure 1bi, iv show the reconstructed virtual dark-field image and charge-density map of the Fe[001]/Fe₃O₄[001] interface, respectively. In the charge-density map (see Fig. 1b-iv), the red and white pixels pinpoint atomic column locations, revealing an atomically sharp Fe[001]/Fe₃O₄[001] interface. The upper part is body-centered cubic (BCC)-Fe. The lower part consists of $Fe_3O_4$, which has an inverse spinel structure characterized by a face-centered cubic (FCC) sublattice of $O^{2-}$ anions, with $Fe^{2+}$ and $Fe^{3+}$ cations occupying the interstitial sites[59]. We observed no abrupt mono-layered transition plane between the two bulk phases; instead, we found a complex reconstruction layer, as characterized by the charge-density map in Fig. 1b–iv. The reconstructed regions four atomic layers away from the interface can be clearly identified in both the upper and lower parts. However, accurately interpreting the atomic structure at the very Fe[001]/Fe₃O₄[001] interface remains challenging when mapping these structural-chemical features only. Hence, in the next section, we derived the underlying complexion diagrams to better understand the reason for this reconstruction.

### The Fe/Fe₃O₄ complexion and its associated complexion diagram

To better understand the structure of the reconstructed Fe[001]/Fe₃O₄[001] interface region, we employed density functional theory (DFT) calculations to search for its thermodynamically most stable configurations. The results are presented in full details in Supplementary Fig. 7. Considering the DFT structural parameters of BCC-Fe and $Fe_3O_4$ (see Supplementary Table 1), the stacking of the two layers results in an in-plane lattice mismatch of $\varepsilon_{xy}^{BCC\text{-}Fe} = -4.5\%$ and $\varepsilon_{xy}^{Fe_3O_4} = +4.7\%$, with respect to the other layer, respectively.

Looking at the observed interface structure of Fig. 1bi, iv, the $Fe_3O_4$ layer appears to adopt a $FeO_2$-termination, and we thus mainly

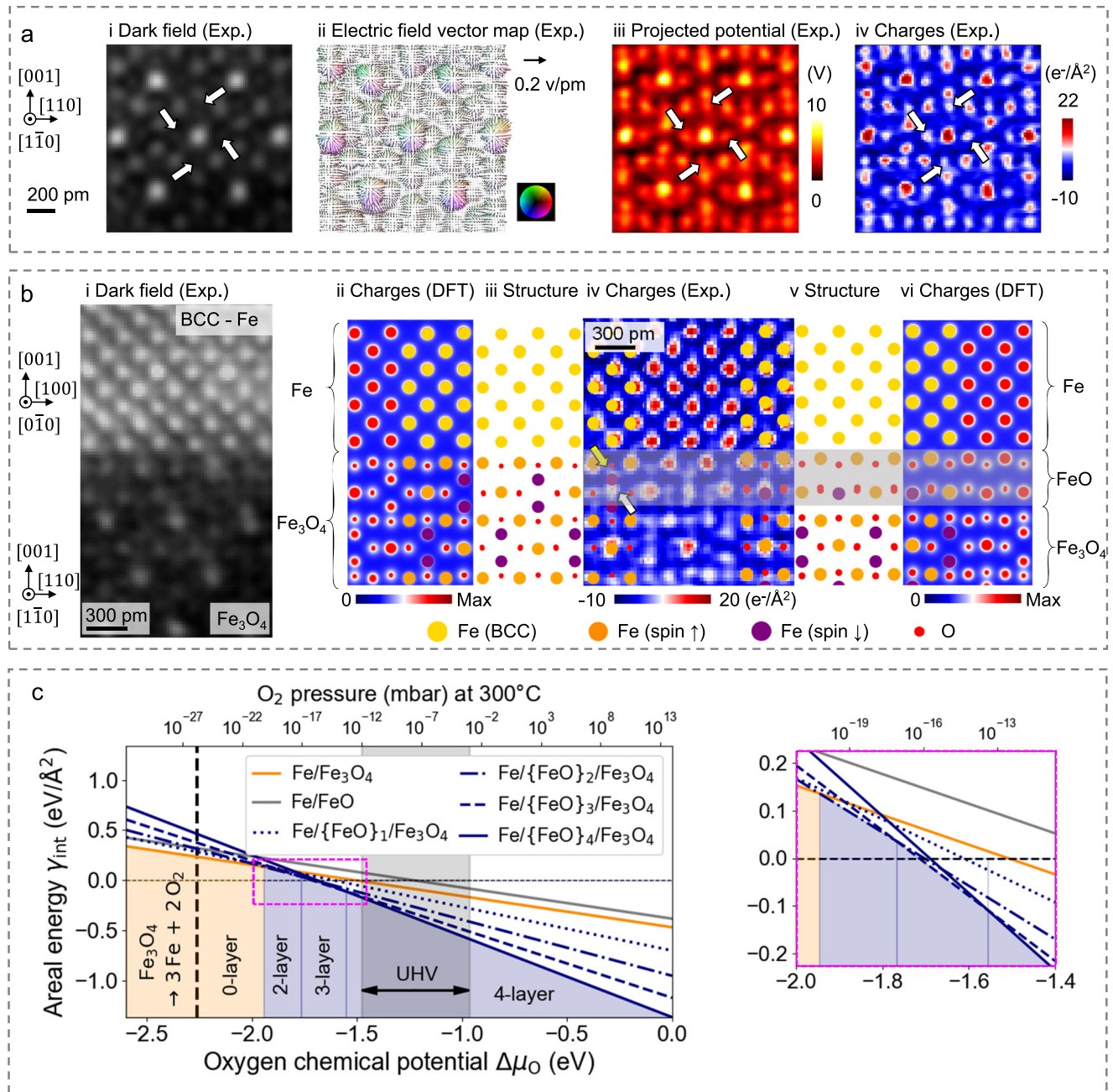

**Fig. 1 | The complexion diagram of the Fe/Fe$_3$O$_4$ interface. a** Experimental (Exp.) differential phase contrast (DPC) - four-dimensional scanning transmission electron microscopy (4DSTEM) reconstruction for Fe$_3$O$_4$[110]: **i** Reconstructed virtual dark-field image **ii** Electric field vector map **iii** Projected electrostatic potential map **iv** Charge-density map. The white arrows in **i, iii**, and **iv** indicate the positions of the O atomic columns. **b** Experimental/theoretical study of the Fe[001]/Fe$_3$O$_4$[001] interface includes: experimental DPC-4DSTEM **i** virtual dark-field image and **iv** charge-density map, along with results from DFT calculations comprising charge-density maps (**ii, vi**) and relaxed structures (**iii, v**) of the interface between Fe[001] and Fe$_3$O$_4$[001]. The first DFT set (**ii, iii**) illustrates the pristine interface configuration, while the second set (**v, vi**) depicts the interface with a reconstructed two-layer thick FeO slab. In **iii, v**, Fe atoms in the body-centered cubic (BCC) structure are represented in yellow. In the Fe$_3$O$_4$ structure, Fe atoms are color-coded based on their spin orientation: spin-up atoms are shown in orange and spin-down atoms in purple. O atoms are depicted in red. In **iii**, yellow and white arrows highlight atomic columns that deviate from the positions observed experimentally in the Fe$_3$O$_4$ structure when the two-layer thick FeO slab is not inserted. **c** DFT predicted complexion diagram versus the O chemical potential shift, with $\Delta\mu_O$ taking molecular O$_2$ as the reference upper limit, and dissociation of Fe$_3$O$_4$ into metallic Fe and molecular O$_2$ as the lower limit (vertical black dashed line). The explored interface configurations include: Fe/Fe$_3$O$_4$ (solid orange), Fe/FeO (solid grey), and Fe/{FeO}$_n$/Fe$_3$O$_4$, with $n$ being the number of FeO layers (dark blue). A horizontal dashed line is included in the graph as a visual reference. The inset on the right shows a magnification in the range of chemical potentials where the transition occurs from four to two intermediate FeO layers, separated by the vertical thin blue line. The O chemical potential is also converted to O$_2$ pressure at the 300 °C temperature of the experiments, with the black shaded region showing typically accessible experimental conditions under ultra-high vacuum (UHV).

focused on this structure in the calculations. We find the lowest energy configuration when BCC-Fe atoms are located at the alternating hollow sites of the upper FeO$_2$-terminated Fe$_3$O$_4$ layer. Here, "hollow sites" refer to positions that are typically in a concave arrangement, surrounded by other atoms, resulting in the strongest adhesion energy of approximately −3 J/m$^2$, representing the energy required to separate the two slabs at their interface into two seperate ones with free surfaces. The calculated charge-density and optimized structure for the

most stable interface configuration generally align well with our experimental observations. This alignment is illustrated in the bridging sketch (see Fig. 1biii), which connects the experimental (see Fig. 1b, iv) and calculated (see Fig. 1bii) charge-density maps. However, there are some discrepancies in certain areas.

Upon closer inspection of the experimental interface structure presented in Fig. 1b–iv, we noted a significant distortion on the Fe$_3$O$_4$ side, particularly within the first two atomic layers. Firstly, in the first atomic layer (closest to the Fe), several O atomic columns (indicated by yellow arrows) appear to have shifted substantially towards the Fe$_3$O$_4$ side. A slight shift of these atomic columns, in the range of a few picometers, is also corroborated by our DFT calculations, as seen in Fig. 1b-ii, but not to a sufficient extent to explain this discrepancy. Secondly, in the second atomic layer (the second one closest to the BCC-Fe), the Fe atoms located at the tetrahedral sites of the FCC-O sublattice (in purple on Fig. 1biv) have noticeably moved away from the Fe side. Simultaneously, the Fe atoms in the fourth atomic layer of the Fe$_3$O$_4$ have shifted towards the Fe side. These two columns of Fe atoms appear to have merged into a single atomic column, as indicated by the white arrows in Fig. 1b–iv.

We propose that the region near the interface adopts a well-ordered two-layer interface-stabilized phase state (referred to as complexion), reminiscent of the iron oxide polymorph FeO (wüstite). FeO exhibits a NaCl structure and becomes thermodynamically stable above 570 °C[60]. In its bulk form, the chemical formula can also be expressed as Fe$_{1-x}$O, where "$x$" represents the degree of Fe deficiency due to Fe vacancies[61–63]. For simplicity, we will subsequently refer to wüstite directly as its stoichiometric form, FeO, yet recall that it is Fe depleted. In the context of the interface, when we now introduce a stoichiometric two-layer FeO slab (complexion) between the Fe and Fe$_3$O$_4$ layers, our DFT calculations align more closely with the experimental results. The difference between the DFT predictions and the observed plane spacing has decreased from 50% to less than 5% with the revised structural model, which includes the FeO slab. This improved correlation is illustrated in the bridging sketch (see Fig. 1b–v), which overlaps the experimental charge-density map (see Fig. 1b–iv) with the theoretical predictions (see Fig. 1b-vi). This effectively resolves the discrepancies previously discussed concerning the O and tetrahedral Fe atomic columns highlighted by yellow and white arrows in Fig. 1b–iv.

To better understand the energetic reason for the occurrence of this FeO-like complexion, we present the relative thermodynamic stability of the different interface structures as a function of the O chemical potential $\Delta\mu_O$ in Fig. 1c (see "Methods" Section Computational Details)[64–66]. For this purpose, we additionally considered the most stable configurations of the Fe[001]/FeO[001] interface and Fe/FeO/Fe$_3$O$_4$ heterostructures comprising 1 to 4 intermediate FeO single atomic layers (see Supplementary Figs. 8–10). Over the whole range of accessible chemical potentials $\Delta\mu_O$, given in terms of the corresponding O$_2$ pressure, the heterostructures are favored according to our analysis. In particular, the number of intermediate FeO layers stabilizing the interface decreases with decreasing pressure or $\Delta\mu_O$. More specifically, a transition occurs from 4 down to 2 intermediate FeO atomic layers at an O chemical potential shift of approximately −1.7 eV. The pristine Fe/Fe$_3$O$_4$ interface is finally favored below an O chemical potential shift of approximately −1.9 eV, corresponding to an O$_2$ pressure lower than $10^{-19}$ mbar at 300 °C, far below the experimental conditions at which the thin films were grown.

We also emphasize that due to the limited system sizes accessible to the DFT calculations, we did not consider more than 4 intermediate FeO atomic layers. However, given the trend observed in Fig. 1c, we expect that the most stable structure in the O-rich region will contain many more FeO layers. A similar effect has been observed recently in DFT calculations of stacked FeO layers on top of a free Fe(001) surface[67], or for grain boundary complexions in a model bicrystal[68].

Also, given the range of accessible chemical potentials in experiments, such extreme cases are not the most relevant ones for the present analysis.

## The effect of complexion formation on the properties of the Fe/Fe$_3$O$_4$ interface

We now investigate in Fig. 2 the impact of the formation of such complexions on the charge transfer at the Fe/Fe$_3$O$_4$ interface using DFT calculations. As can be inferred from the change in the structure of the whole interface, the transfer properties are also impacted by the presence of FeO-like complexions. In both cases (with and without a complexion present), most of the charge transfer occurs within the inter-layer region, with electrons coming from the oxide layer. We also note a stronger electronic transfer at the interface between FeO and BCC-Fe (Fig. 2c, right panel), which is not present at the clean binary Fe/Fe$_3$O$_4$ interface (Fig. 2a, b). This is an indication of an increased strength of the whole interface in the case of the Fe/{FeO}$_4$/Fe$_3$O$_4$ heterostructure. This is supported by the stronger predicted work of adhesion of −3.9 J/m$^2$ between iron and the oxide for the heterostructure compared to −2.7 J/m$^2$ for the binary interface (see Supplementary Table 2). This is caused by the strong adhesion between BCC-Fe and FeO, also in line with other DFT studies on the Fe/FeO interface[66,67].

We also investigate the influence of the magnetic coupling between the BCC-Fe and the oxide layer on the electronic transfer, focusing in Fig. 2A, B on the clean binary Fe/Fe$_3$O$_4$ interface. A ferromagnetic (FM) coupling between the two layers is energetically more favourable. However, the energy difference between the FM and antiferromagnetic (AF) couplings amounts to only 0.1 J/m$^2$, qualifying the two types of magnetic coupling at the interface almost as degenerate states. On the other hand, the behavior of the charge transfer at the very interface changes quite noticeably from one magnetic coupling to the other. In particular, the negative transfer is more localized with an AF coupling, centered around the position of the outermost octahedral Fe atom of the Fe$_3$O$_4$ layer.

Finally and most importantly, the change in the structure of the Fe/Fe$_3$O$_4$ interface with the oxygen chemical potential, driving the formation of FeO complexions of various thicknesses, will greatly impact the transport properties of a material containing these defects. Indeed, wüstite is inherently off-stoichiometric, with a depleted iron sub-lattice. Thus the presence of FeO layers at the Fe/Fe$_3$O$_4$ interface will act as a sink for iron vacancies. These interfaces are omnipresent during redox processes involving iron and its oxides, for instance during hydrogen-based reduction of oxides[27], or during the combustion of iron powder[69]. Thus, an understanding of their structure is of great importance for modeling the transport of atomic species during these processes.

## Formation of the Fe/Fe$_3$O$_4$ complexions independent of the processing path

Experimentally, we have employed atmospherically controlled PVD to produce the specimens in which we examine the Fe/Fe$_3$O$_4$ interface under varying levels of external O activity (an equivalent measure of chemical potential, controlled by the O$_2$/Ar ratio and pressure). This is illustrated in Fig. 3a and further demonstrated by energy-dispersive X-ray spectroscopy (EDS) mapping in Fig. 3b. Our aim was to create multi-layer thin films where the Fe/Fe$_3$O$_4$ interfaces occur under different conditions. This enables us to assess the general applicability of our observations regarding interfacial structures associated with FeO. In one setup, BCC-Fe was deposited on Fe$_3$O$_4$ under conditions of very low external O activity, as shown in Fig. 3c and Supplementary Fig. 6. In contrast, under a high O activity environment, Fe$_3$O$_4$ was deposited on BCC-Fe, as depicted in Fig. 3d and Supplementary Fig. 11. It should be noted that FeO-like complexions were observed in both scenarios, as indicated by the arrows in Fig. 3c, d.

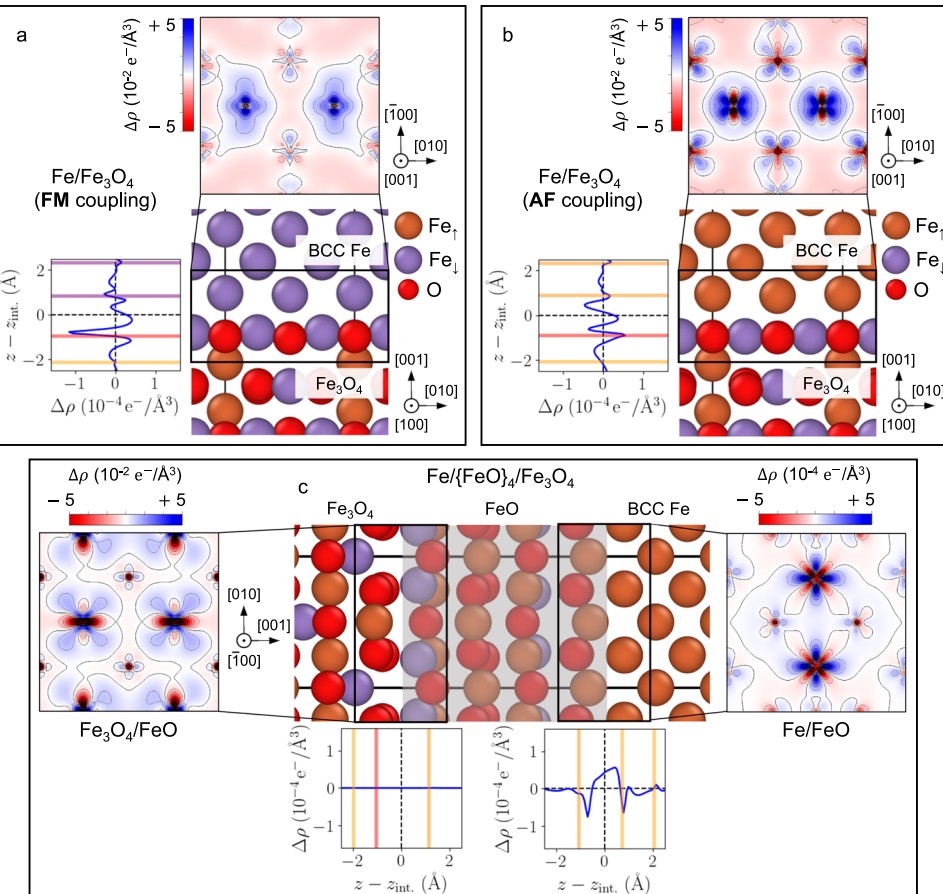

**Fig. 2 | Charge transfer at the Fe/Fe₃O₄ interface and the effect of the presence of FeO-like complexion. a** Charge transfer at the Fe/Fe₃O₄ interface when the body-centered cubic (BCC)-Fe layer is coupled ferromagnetically (FM) to the outermost FeO₂-terminated Fe₃O₄ layer (all Fe spins pointing down), and **b** with the BCC-Fe layer antiferromagnetically (AF) coupled to Fe₃O₄. **c** Charge transfer when a four-layer FeO complexion is located between BCC-Fe and Fe₃O₄, both at the FeO/ Fe₃O₄ (left) and the BCC-Fe/FeO (right) interfaces. For each subplot is shown the in-plane projection of the charge transfer $\Delta\rho$ in the vicinity of the interface and the line profile in the direction normal to the interface plane as a function of the distance $z - z_{int.}$ to the interface height $z_{int.}$. Crystallographic orientations are given with respect to the BCC-Fe lattice.

Our multi-layer deposition synthesis approach serves as a method to examine the interface contribution in stabilizing such complexions or any related interfacial phase states. The thermodynamic discussion and further details on the formation of bulk Fe and Fe₃O₄ phases during thin film deposition can be found in the "Methods" Section Computational Details and Supplementary Figs. 12 and 13. Our observations at the interface between Fe and Fe₃O₄ involve an interfacial structure closely tied to the FeO structure. In both cases, Fe on Fe₃O₄ or Fe₃O₄ on Fe, FeO-like complexions are stabilized at the interfaces. Interestingly, our experimental conditions are based on a synthesis at 300 °C under constant external O activity, where FeO, a high-temperature stable oxide, exists as a interfacial phase under ambient pressure above 570 °C[60]. Such observations of FeO-like complexions occur when the interface evolves towards a complexion-mediated equilibrium state. Our findings demonstrate a pathway for stabilizing interface phase states that are not stable according to the bulk phase diagram but can be made thermodynamically stable when they are between two adjacent phases, through a deliberately designed interface structure.

### Diversity of Fe/Fe₃O₄ interfacial structures

Throughout our discussion, we have primarily focused on the observation of a two-layer FeO-like complexion at the Fe[001]/Fe₃O₄[001] interface. Our DFT calculations also predict the stability of thicker FeO-like complexions, as shown in Fig. 1e in the O-rich region. Additional

observations in Fig. 4, Supplementary Figs. 14 and 15 corroborate this prediction, showing that in certain regions of the Fe/Fe₃O₄ interface, the three-layer and four-layer FeO-like complexion are evident, respectively. Such heterogeneity in complexion thickness may be attributed to a local fluctuation of synthesis conditions leading to a local O₂ activity variation. As seen in Fig. 1c, the configurations of the FeO-like complexion found at the Fe/Fe₃O₄ interface demonstrate a dependence on local O chemical potential.

It is also worth mentioning that the local strain build-up due to the mismatch between adjacent phases can be another factor in controlling the interface complexions. While the lattice mismatch between Fe and Fe₃O₄ is relatively small, this is not the case with FeO, with $\varepsilon_{xy}^{BCC\text{-}Fe} = -6.8\%$ and $\varepsilon_{xy}^{FeO} = +7.3\%$ with respect to the other layer. This mismatch also favors the formation of defects, such as misfit dislocations and steps, at the Fe/Fe₃O₄ interface to alleviate strain, examples of which are illustrated in Supplementary Figs. 3, 16 and 17. For the regions without misfit dislocations, we measured lattice parameters for the Fe phase, the FeO-like interfacial phase, and the Fe₃O₄ phase to be approximately 0.29 nm, 0.43 nm, and 0.83 nm, respectively (see black lines in Supplementary Fig. 18). These measurements are consistent with our DFT-predicted values of 0.28 nm, 0.43 nm, and 0.84 nm, respectively, confirming the existence of the interfacial FeO-like phase.

In summary, in this study, we employed a combination of advanced characterization techniques and DFT calculations to explore

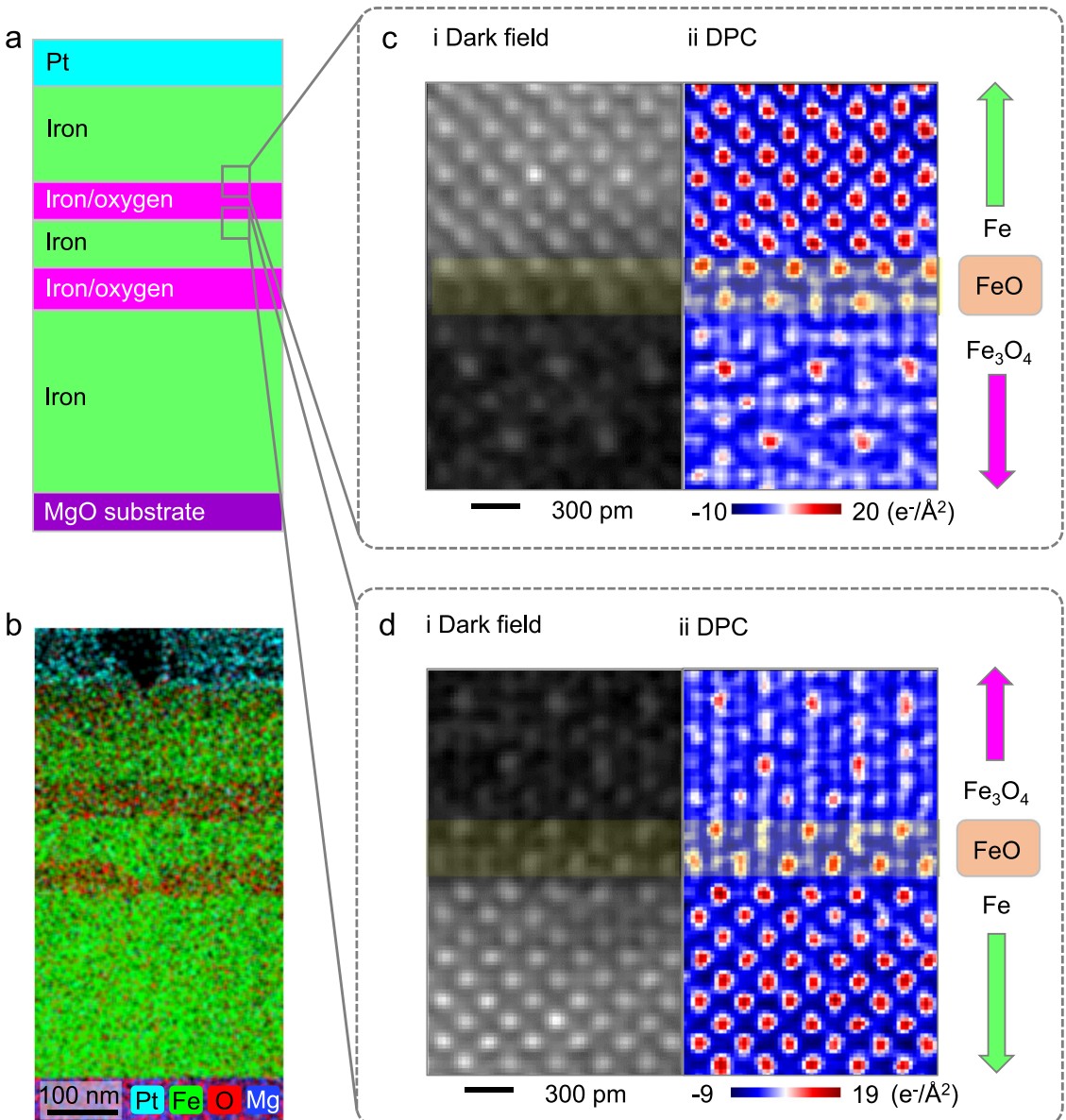

**Fig. 3 | The Fe/Fe₃O₄ complexions form under various fabrication conditions.** **a** Multi-layer thin film deposition under different controlled external atmospheres, with or without flowing $O_2$ gas. **b** Energy-dispersive X-ray spectroscopy (EDS) map showing the cross-section view of the thin film. **c** Fe is depositing on $Fe_3O_4$ under a pure Ar atmosphere, where the FeO-like complexion is observed at the interface. **d** $Fe_3O_4$ is depositing on Fe under a mixed $Ar/O_2$ atmosphere, where the FeO-like complexion is also observed at the interface. In (**c**, **d**) the areas between the red arrows indicate the regions of FeO-like complexions. DPC: differential phase contrast imaging, conducted in a scanning transmission electron microscope.

the structural and chemical features at the interfaces between iron and its oxides. Experimentally, we observed interfacial FeO-like complexions at the Fe[001]/Fe₃O₄[001] interface and used DFT-based thermodynamics to demonstrate their higher stability compared to an interface without presence of such complexions. Our findings provide an explanation for the formation of complexions at the phase boundaries. Using DFT calculations, we constructed the complexion diagram of the interface, explaining the origin of the different types of observed interface phase states, depending on the oxygen chemical potential and the adjacent bulk phases. The atomic structure of the complexions we observed appears in specific phase states, which are significantly different from their equilibrium bulk existence and impact the electrical, mechanical, and transport properties of the interface. This study thus lays the groundwork for the possible future integration of such structural-chemical complexion states as a material and process design toolbox, accessible to crisp thermodynamics rules.

The specific defect phase state we studied here can leverage profound kinetic, thermodynamic, and mechanical effects that can alter mass and charge transport, phase transformation behavior, interface strength and catalysis, to name but a few features. This highlights the broader principle that thermodynamically stable interface structures can be predicted, designed, implemented and functionally utilized with significant opportunities for advanced materials, processes and interface-related reaction phenomena.

## Methods
### Synthesis
Five-layered Fe/iron oxide thin films were prepared in a BesTeck PVD cluster (MPIE, Düsseldorf, Germany) by direct current (DC) sputtering a pure Fe target (99.99%, Mateck, Germany) with a power of 150 W in the alternated Ar and $O_2$ gas environment at 300 °C. The chamber was pumped to a base pressure of $5.0 \times 10^{-8}$ mbar before the sputtering

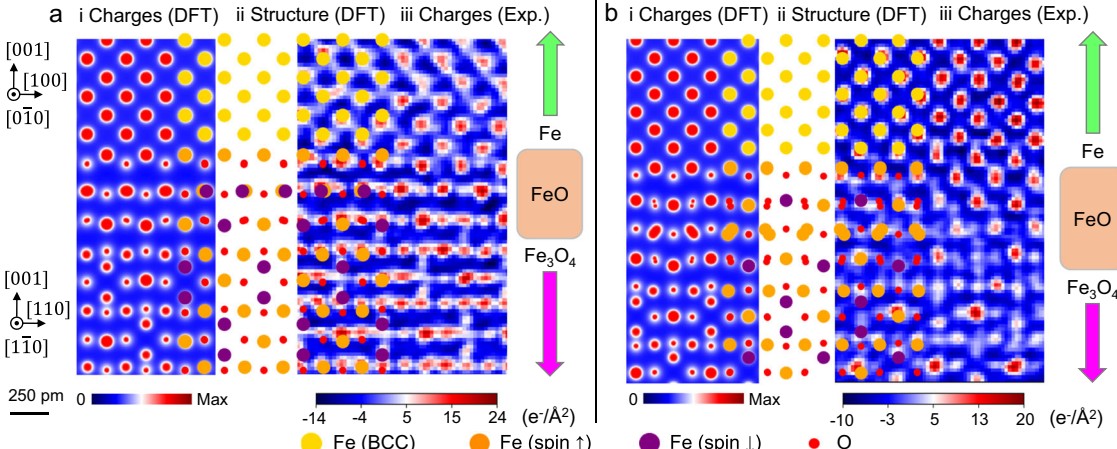

**Fig. 4 | Different Regions of the Fe/Fe₃O₄ interface displaying FeO-like complexions of various thicknesses. i** DFT-computed electronic charge densities, **ii** DFT-relaxed equilibrium structure, and **iii** DPC-4DSTEM reconstructed charge density map for **a** three-layer and **b** four-layer FeO-like complexions at the Fe/Fe₃O₄ interface. The relaxed DFT structure of the interface is presented as a bridging sketch between the two charge density maps in both (**a**, **b**). DFT: Density functional theory. DPC: differential phase contrast imaging, conducted in a scanning transmission electron microscope.

experiments. Subsequently, the Fe thin film layer was first deposited on a [001] textured single-crystalline MgO substrate at a pressure of $5 \times 10^{-3}$ mbar and an Ar flux of 40 standard cubic centimeters per minute (SCCM). Then an extra $O_2$ flux of 2 SCCM was introduced for sputtering the iron oxide thin film layer. The shutters for both the Fe target and the substrate were closed during the transition of the sputtering gas environment. The deposition rate was kept at approximately 1.9 nm/min. By modulating the gas environment, we obtained a multi-layer thin film with a total thickness of approximately 260 nm, see Supplementary Fig. 1a. After sputter deposition, the thin film was then cooled to room temperature in the vacuum chamber.

### Characterization

**Electron microscopy.** We prepared the transmission electron microscopy (TEM) lamellae using the lift-out procedure in an FEI Helios Nanolab 600i focused ion beam (FIB) dual-beam microscope. Initially, the TEM lamellae were thinned to a thickness of less than 50 nm using an accelerating voltage of 30 kV. Subsequently, a careful polishing step was performed at an accelerating voltage of 5 kV. The thinnest region of the TEM lamellae can reach a thickness of less than 10 nm. The STEM-HAADF images and EDS data presented in Supplementary Figs. 1b&c were acquired using a Cs probe-corrected FEI Titan Themis 60–300 microscope operating at 300 kV. The semi-convergence angle was set to 23.6 mrad, and the semi-collection angle ranged from 78 to 200 mrad. The EDS data was collected by acquiring a series of at least 500 frames, with each probe dwell time recorded for a duration of 20 $\mu$s. The interface between Fe and iron oxide is sharp, but exhibits a faceted morphology that can be seen in the STEM-HAADF image (see Supplementary Fig. 1b) and the EDS mapping (see Supplementary Fig. 1c). In this work, we specifically study the defect-free and strain-free interface to isolate factors influencing complexion formation. This clean interface structure (see Figs. 1 and 3) not only facilitates accurate high-resolution imaging and DPC measurements but also enables direct comparisons with DFT calculations to provide deep insights into the atomic mechanisms at play.

We analyzed the phase and orientation of the multi-layer thin films using the precession-assisted 4DSTEM technique[70], as shown in Supplementary Fig. 2. The 4DSTEM data sets were acquired using the TemCam-XF416 pixelated complementary metal-oxide-semiconductor detector (TVIPS) installed in a JEM-2200FS TEM (JEOL) operating at 200 kV. To create a quasi-kinematic diffraction pattern, we precessed the incident electron beam by 0.5° while scanning the specimens with a step

size of 3 nm. The collected 4DSTEM data set was then indexed using the ASTAR INDEX program, and the phase and orientation were mapped using the TSL OIM Analysis 8 software package. Structural characterization in Supplementary Fig. 2a reveals that the dominant phases for the Fe layers and the oxide layers are the BCC phase and the inverse spinel phase, respectively. The primary out-of-plane growth directions for both phases appear to be [001] (see Supplementary Fig. 2b), suggesting an epitaxial growth mechanism. Supplementary Fig. 2c presents the in-plane orientation between different layers. A 45° rotation has been observed between the BCC-Fe and the Fe₃O₄ layers, indicating an orientation relationship matching the one determined previously by Davenport et al.[71], namely Fe[001]$_z$||Fe₃O₄[001]$_z$, Fe[100]$_x$||Fe₃O₄[110]$_x$.

The DPC-4DSTEM data sets were acquired using the electron microscope pixel array detector (EMPAD) equipped in the aforementioned Titan microscope at 300 kV. For each probe position, the EMPAD captured the convergent beam electron diffraction (CBED) pattern at a semi-convergence angle of 23.6 mrad, with an exposure time of 1 ms per frame. All CBED patterns had a uniform size of $128 \times 128$ pixel² with a pixel size of 2.0 mrad. The scanning step size was selected within the range of 13–25 pm. A smaller scanning step size offers a higher spatial resolution for accurately resolving the positions of atomic columns. This is demonstrated in Supplementary Fig. 4, which showcases the experimental DPC-4DSTEM reconstruction of Fe₃O₄ oriented in the [110] direction. However, it is important to note that a smaller scanning step size can introduce distortions due to sample drifting. On the other hand, scanning with a larger step size can reduce image distortion, but it may compromise the resolution required to resolve lighter O atomic columns. After optimization, we determined that the optimal scanning step size for the current experiments is 18 pm, as depicted in Supplementary Fig. 5. Annular bright field (ABF)-STEM imaging concurrently provides contrast for both heavy Fe atomic columns and light O atomic columns[72–76]. However, ABF-STEM phase contrast measurements are challenging due to complex contrast transfer functions[77]. In this study, we consistently employed DPC-4DSTEM as the primary method for imaging the interface structure as we find the multi-faceted information it provides to be valuable to help us address our scientific questions effectively.

We reconstructed the DPC-4DSTEM data sets using the in-house developed Python script pyDPC4D (GitHub link: https://github.com/RhettZhou/pyDPC4D). The reconstructed data provide information including the virtual dark-field image, the center of mass of the transmitted beam, the electric field vector map, the projected

electrostatic potential map, and the charge-density map, see Supplementary Fig. 5. Here we will briefly review the key steps of the data reconstruction. More details regarding the data reconstruction are referred to our previous publication[58].

The DPC-4DSTEM data set consists of a two-dimensional (2D) grid of probe positions in real space along with corresponding 2D diffraction patterns in reciprocal space.[56,78] Each recorded diffraction pattern exhibits a bright field disk that contains information about the momentum transfer of the electron beam[51–55]. By analyzing the displacement of the center of mass of the bright field disk, we can track the momentum transfer of the electron beam as a function of the probe position. According to the Ehrenfest theorem[79], the electric field of a thin specimen is related to the momentum transfer of the electrons in the beam[80]. The integration of the electric field yields the projected electrostatic potential[52–54]. The charge-density (including electrons and protons) is proportional to the divergence of the electric field as dictated by Gauss's law[52,53,55].

We conducted in situ heating experiments to examine the stability of the FeO-like complexion. Using the lift-out procedure with an FEI Helios Nanolab 600i FIB dual-beam microscope, we prepared the TEM lamellae on a DENSsolutions chip. We then performed electron microscopy experiments in the Cs probe-corrected FEI Titan Themis 60−300 microscope using similar operating parameters as previously mentioned. The experimental conditions included 2 h of in situ TEM heating at 300 °C, with a heating and cooling rate of 5 °C/s. Overall, the structure remained unchanged after prolonged heating; although we observed some local reorganization through a ledge growth mechanism at the atomic scale, the FeO-like complexion consistently appeared at the interface (see Supplementary Figs. 14 and 19).

For quantifying local composition and valence state transitions across the interface, we used a Gatan Quantum spectrometer to acquire electron energy loss spectroscopy (EELS) data at 300 kV with an entrance aperture of 35 mrad. We estimated the atomic ratio of Fe to O by integrating the intensities of the Fe-L3 and O-K edges. Additionally, we performed peak decomposition on the Fe-$L_3$ edges to determine the fractions of various oxidation states[81]. As Supplementary Fig. 20 illustrates, EELS analysis provides insights into the O atomic ratio and valence state transitions across the interface of the specimen heated in situ at 300 °C for 2 h. Although the complexion region appears in HAADF images, a gradient in valence states at the interface, rather than a sharp transition, occurs due to beam broadening and overlapping signals of Fe, $Fe^{2+}$, and $Fe^{3+}$ in $Fe_3O_4$, complicating the direct observation of complexions using EELS.

**Multi-slice image simulation.** We generated a synthetic 4D-STEM data set using the μSTEM simulation suite, which employs the multislice method[82]. By comparing the experimental and simulated CBED patterns (see Supplementary Fig. 21), we estimated the specimen thickness to be approximately 8.3 nm. We analyzed the simulated data set following the same procedure as that for the experimental data in the DPC-4DSTEM analysis[58]. The simulation setup was intended to mimic the parameters used during experimental data acquisition, such as an accelerating voltage of 300 kV, a semi-convergence angle of 23.6 mrad, a sample thickness of 8.3 nm, and a probe spacing of 13 pm. Supplementary Fig. 22 displays the reconstruction for $Fe_3O_4$ oriented in the [110] direction. To elucidate the origin of the contrast observed in the charge-density map, we conducted three series of simulations, varying parameters such as sample thickness, focus, and probe spacing, as illustrated in Supplementary Fig. 23.

## Computational Details

**DFT calculations.** All ab initio calculations reported in this work were performed within DFT, as implemented in the VASP code[83]. Projector augmented wave (PAW) pseudopotentials including 8 and 6 valence electrons were used to model Fe and O atoms respectively, with the exchange-correlation potential approximated using the GGA-PBE functional[84]. A plane-wave basis at a cutoff energy of 500 eV was used, and the Brillouin zone was sampled by a $\Gamma$-centered $k$-point mesh of density $0.02 \, 2 \, \pi \, Å^{-1}$.

Magnetism is treated within the collinear approximation. For the BCC-Fe and $Fe_3O_4$ layers, all magnetic moments are initialized as in the ground-state magnetic order of each structure, namely ferromagnetic (FM) for BCC-Fe and ferrimagnetic (FeM) for $Fe_3O_4$. As for the FeO-like complexions, we search - by using DFT - for the lowest energy magnetic ordering state by considering different arrangement of the Fe magnetic moments within the complexion layer. We did so since it might well be that the bulk AF order of FeO is not the most energetically favorable, given the thinness of the complexions. Only the lowest energy configurations are presented, and all values derived from these structures, which all differ from the bulk AF magnetic order of FeO.

Structural parameters (lattice parameter and bulk modulus) for the three materials considered here (BCC-Fe, FeO and $Fe_3O_4$) are presented in Supplementary Table 1, computed using both standard DFT (GGA-PBE functional) and DFT + $U$, and compared to experimental references. It is worth noting that standard DFT fails to correctly account for the electronic correlations found among iron oxides, therefore usually motivating the use of a Hubbard $U$ term on 3$d$-orbitals of Fe atoms in the framework of DFT + $U$. However, such a correction yields a poor description of pure Fe (as can be seen in Supplementary Table 1) for structural parameters, but also the relative stability between different crystal structures and magnetic orders. Since both pure Fe and iron oxides coexist in all interface structures studied here, we chose to use a consistent description of Fe atoms across all three materials, therefore opting for standard DFT, without Hubbard $U$ correction. This choice yields structural properties in good agreement with experimental data for all three materials (see Supplementary Table 1).

Charge-density plots (see for instance Fig. 1b, d) were obtained from "all-electron" DFT calculations in the sense of the PAW method, *i.e.* as the sum of the self-consistent valence electron density and the core density contained in the PAW pseudopotentials.

Electron transfer at the interface, presented in Fig. 2, is obtained through the DFT-computed electronic charge density difference $\Delta\rho$ defined as

$$\Delta\rho = \rho_{int.} - \sum_i \rho_{slab, i} \tag{1}$$

where $\rho_{int.}$ is the charge density of the whole interface, and $\rho_{slab,i}$ are the charge densities of the different slabs contained in the interface model, *i.e.* for the binary BCC-Fe/$Fe_3O_4$ the two BCC-Fe and the $Fe_3O_4$ slabs, and for the heterostructure, the BCC-Fe, the FeO layers as a whole and the $Fe_3O_4$ slabs. The projection in the $xy$-plane is obtained by integrating the charge density difference in the volume contained across the area of the interface, with $\pm 2$ Å height from the location of the interface. Line profiles were obtained by integrating the charge density difference in the $xy$-plane at different heights along the direction normal to the interface plane.

**Interface models.** All interface models are constructed with periodic boundary conditions along all three Cartesian directions. They contain stackings of 7 layers of BCC-Fe(100) planes, 9 layers of $Fe_3O_4$(100) planes, and up to 5 layers of (100) FeO planes. Initial lattice mismatches between the different materials in all interfaces considered are presented in Supplementary Table 2. When constructing the interfaces, the in-plane lattice parameter is fixed, while the lattice parameter is kept to its bulk value for each slab along the direction normal to the interface plane.

For all interface configurations, we then start by varying the relative position of the two slabs specifically among the high-symmetry

sites. The adhesion energy $\gamma_{adh}$ between the two slabs of the interface is then determined as:

$$\gamma_{adh} = \frac{E^{tot} - \sum_i E_i^{slab}}{2\,S}, \qquad (2)$$

where $E^{tot}$ is the total energy of the interface model, $E_i^{slab}$ are energies of the individual slabs in vacuum, and $S$ is the in-plane area of the interface. Using this definition, a negative adhesion indicates a stable interface between slabs in presence, and allows to find its most stable configuration.

For each site, $\gamma_{adh}$ is then evaluated as a function of the interface separation distance $d_{surf}$ without relaxing atomic positions (see for instance Supplementary Fig. 7 for the case of the Fe[001]/Fe$_3$O$_4$[001] interface). For the most stable site among the ones considered, the atomic positions are then allowed to relax until the remaining forces on all atoms in all three Cartesian directions are less than 5 meV/Å. We also allowed the geometry of the cell to relax until the remaining stresses are close to zero. The strains after relaxation are given for each layer in Supplementary Table 2 for all interfaces considered.

The structure presented in Fig. 1d–i, and all Fe/FeO/Fe$_3$O$_4$ heterostructures, are obtained by matching the most stable relative positions of the three BCC-Fe, FeO and Fe$_3$O$_4$ layers for each single interface structure, i.e. between Fe/FeO and FeO/Fe$_3$O$_4$. Looking at the heterostructures, this results in a slight change of the most stable position of the BCC-Fe slab with respect to the Fe$_3$O$_4$ slab as compared to the Fe/Fe$_3$O$_4$ interface of Fig. 1b–ii. This can be rationalized considering the proximity between the two FeO and Fe$_3$O$_4$ structures. Indeed, they share the same FCC-O sublattice, while the Fe atoms fill different interstitial sites for the two oxide structures. Here, the structure of the interface presented in Fig. 1d–i, which is the most stable configuration, corresponds to the position of the FeO layer with respect to the Fe$_3$O$_4$ layer where these two FCC-O sublattices match.

**Complexion diagram.** The complexion diagram presented in Fig. 1e is constructed as detailed in refs. [64,65]. For this purpose, we need the interface energy for each structure of interest, computed using DFT. Those are Fe/Fe$_3$O$_4$, Fe/FeO and Fe/(FeO)$_n$/Fe$_3$O$_4$, with $n$ the number of intermediate FeO layers intercalated between the BCC-Fe and Fe$_3$O$_4$ slabs.

To study the stability of each of these structures, we model the experimental conditions as the following: a clean Fe(001) surface (deposited on the MgO substrate) is then covered with Fe and O atoms coming from reservoirs of given chemical potentials $\mu_i$. The chemical potentials of each species are given by $\mu_{Fe} = E_{Fe}^{BCC} + \Delta\mu_{Fe}$ and $\mu_O = 1/2 E_{O_2}^{O_2} + \Delta\mu_O$, where $E_{Fe}^{BCC}$ and $E_{O_2}^{O_2}$ are the DFT energies of BCC-Fe and the O$_2$ molecule respectively, while $\Delta\mu_{Fe}$ and $\Delta\mu_O$ are the changes in their chemical potentials with respect to these two reference states. The relative energies $\gamma$ of the different interface configurations, defined per surface area, are then given by:

$$2\,S\,\gamma = \left[ E_{int.}^{tot} - E_{ref.} - n_{Fe}\,E_{Fe}^{BCC} - \frac{1}{2} n_O\,E_{O_2}^{O_2} \right] - n_{Fe}\,\Delta\mu_{Fe} - n_O\,\Delta\mu_O \qquad (3)$$

with $E_{int.}^{tot}$ the total energy of the interface, $E_{ref.}$ the energy of the reference (here a clean Fe(001) surface), $n_{Fe}$ and $n_O$ the number of Fe and O atoms contained in the deposited slabs (i.e. excluding the reference Fe(001) surface). For the analysis presented in Fig. 1e, we set $\Delta\mu_{Fe} = 0$ and let the change in O chemical potential $\Delta\mu_O$ be the varying parameter. Also, the in-plane lattice parameter of each material (BCC-Fe, FeO and Fe$_3$O$_4$) is set to the DFT-computed one of BCC-Fe, to mimic the experimental conditions where the oxide layer is deposited on top of a BCC-Fe layer. The same complexion diagram was also computed using the lattice constant of Fe$_3$O$_4$ to mimic the other experimental condition. A third set of calculations was also carried using the average

lattice parameter of the three materials, weighed by their respective bulk moduli. The exact same trend (i.e. transition from a 4-layer FeO complexion to the clean Fe/Fe$_3$O$_4$ interface with decreasing oxygen chemical potential) was observed under all three conditions.

**Bulk thermodynamics.** The Fe/Fe$_3$O$_4$ multilayer thin film is created by atmospherically controlled PVD under varying levels of external O activity, and this is compared with bulk thermodynamics, which serves as a guide for thin film deposition. It should be emphasized that the bulk thermodynamics conducted here are not aimed at explaining the kinetic path of interface complexion formation, but rather at aiding the understanding of bulk phase formation during thin film deposition.

We compared two types of Gibbs energy evaluations. The first type involves full equilibrium calculations, mapping out stable phases as functions of both temperature and O activity. Supplementary Fig. 12a shows the thermodynamic equilibrium as a function of O activity and temperature. At high O activity, this equilibrium favors Fe$_2$O$_3$ (hematite). Notably, FeO is absent across the neighboring regimes of the synthesis conditions illustrated in Supplementary Fig. 12a.

The second type focuses on the instantaneous driving force at the onset of the reaction, which represents the energetic shift in the system when an infinitesimal amount of solid phase (either Fe or iron oxide) is deposited from a supersaturated gas phase[85]. A metastable phase is often frozen during low-temperature deposition due to the limited diffusivity for further phase transformation. Therefore, we computed the instantaneous chemical driving force for reaction onset for iron oxide polymorphs and BCC-Fe, in reference to depositing from the gas phase, as a function of temperature and O activity.

In Supplementary Fig. 12b, we labeled the plots with O activities corresponding to the experimental conditions: 1) BCC-Fe-forming condition at an infinitesimal O activity at 300 °C; 2) an experimental magnetite-forming condition, which corresponds to a volumetric flow rate ratio of 40/2 for Ar and O at 300 °C; 3) an experimental Fe$_2$O$_3$-forming condition, which corresponds to a volumetric flow rate ratio of 40/10 for argon and O at 25 °C. All PVD experiments were conducted at a pressure of 0.5 Pa ($5 \times 10^{-3}$ mbar). We observed a consistent match between the phases synthesized under specific conditions (see Fig. 3c, d) and the thermodynamic predictions derived from calculating the maximum instantaneous chemical driving force.

We have used the Gibbs energy assessments of BCC-Fe from the TCFE13 database, as well as Fe$_2$O$_3$, Fe$_3$O$_4$, and FeO, and the gas phase, from the SSUB5 database of Thermo-Calc. Additional details can be found in the GitHub repository for these calculations: https://github.com/YXWU2014/IronOxide_TC[86].

## Data availability

The data generated in this study have been deposited in the public community repository Figshare: https://doi.org/10.6084/m9.figshare.28342463.

## Code availability

The Python code used for the electron microscopy analysis in this study is available on GitHub: https://github.com/RhettZhou/pyDPC4D. The Python code used for the Thermo-Calc study is also available on GitHub: https://github.com/YXWU2014/IronOxide_TC.

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

## Acknowledgements

The authors highly appreciate the discussions with Prof. Dr. Jörg Neugebauer, Prof. Dr. Gerhard Dehm, Prof. Dr. Christina Scheu, Prof. Dr. Baptiste Gault, Dr. Mira Todorova, Dr. Christoph Freysoldt, and Dr. Siyuan Zhang. D.R. acknowledges funding by the European Research Council (ERC) via the ERC advanced grant 101054368. Views and opinions expressed are however those of the authors only and do not necessarily reflect those of the European Union the ERC. Neither the European Union nor the granting authority can be held responsible for them. X.Z. acknowledges funding by the German Research Foundation (DFG) through the project HE 7225/11-1. B.B. acknowledges support from Alexander von Humboldt Foundation. C.O. from the Molecular Foundry acknowledges support from the Office of Science, Office of Basic Energy Sciences, of the U.S. Department of Energy under Contract No. DE-AC02-05CH11231.

## Author contributions

D.R. and X.Z. secured funding. D.R. conceived of the presented idea and supervised the project. X.Z. conducted the experimental study. B.B. performed the atomistic simulations. Y.W. and A.K. executed the thermodynamic calculations. C.O. contributed to the electron microscopy data analysis. X.Z., B.B., and Y.W. wrote the original paper. All the authors revised the paper.

## Funding

## Competing interests

The authors declare no competing interests.
