## [Transparent Peer Review file · Nature Communications]

Complexions at the Iron-Magnetite Interface

Corresponding Author: Dr Xuyang Zhou

Version 0:

Reviewer comments:

Reviewer #1

(Remarks to the Author)

The submission titled "Complexions at the Iron-Magnetite Interface" discusses an experimental and computational study on the formation and stability of phase boundary complexions between iron and iron oxide in multilayer films. The key observation was that phase boundary complexions between Fe and Fe oxides in a PVD thin film structurally transitioned as a function of oxygen chemical potential. The results were supported by density functional theory calculations, and the potential impact on bulk material properties was presented. This was a well-written manuscript with exemplary characterization results.

Detailed comments. Comments 3-6 need to be addressed:

1. The introduction nicely reviews the importance of bulk oxide materials and their interfaces, and sufficiently introduces the framework of grain boundary complexions to conceptualize interfacial transformations and properties. The necessity of advanced electron microscopy techniques to study complexions in oxide systems was all well formulated.
2. The experimental results are excellent, particularly the use of differential phase contrast (DPC) to image the positions of oxygen-rich columns, as well as extracting projected potentials to most adequately compare to the density functional theory calculations.
3. A major source of confusion in the manuscript is the inconsistent use of terminology. The literature comprises of two nomenclatures that are intended to describe to the same phenomena, namely grain boundary 'complexions' and grain boundary 'phases'. Based upon the title and the rationale discussed by Cantwell et al in their comprehensive review paper on grain boundary complexions we recommend that the authors use the complexion terminology. For example, the section titled 'The Fe/Fe₃O₄ complexion and its associated interfacial phase diagram' contains both grain boundary complexions and phases. An option would be to call the thermodynamic diagrams as 'complexion diagrams'.
4. The main issue in our opinion is whether the complexions observed are in equilibrium, and thus the stable complexion type. In the reviewer's opinion, the authors should provide evidence or a strong argument that the 'multilayer of FeO' is a complexion rather than a bulk phase that 'precipitates' on the phase boundary. The results currently only show as-deposited films. Long term heat treatments should be performed to assess the complexion stability and if the structure observed currently is maintained (or different). Another source of confusion is that the authors use FeO to describe the complexion, which is a bulk phase. The authors should not use dashed lines to draw both sides of the complexion. A complexion is the interface itself (one interface) as compared to a wetting film that is comprised of two newly created interfaces (phase boundaries). The DFT work is suspected to artificially reinforce the notion that the observed interfaces are composed of a bulk phase at the interface since FeO serves as a transition 'zone' in valence state of Fe.
5. The atom probe tomography data is inconsistent with the conclusions that the phase boundary complexion is uniform despite the growing surface. Figure 3 claims that the complexion is identical if the Iron/Oxygen layer grows on the Iron layer and vice versa. However, the atom probe tomography shown in Supplemental Figure 15 shows a shoulder on one interface. Is the atom probe tomography data necessary?
6. Supplemental Figure 3 shows HAADF micrographs of the iron | iron/oxygen interface, but there is no explanation on why these interfaces are non-planar. How does this data compare to the 4D STEM results shown in Figure 1?
7. Finally, while bulk material properties were discussed in the introduction, there is a lack of physical measurements in this work. Without measurements of property data (mechanical properties, diffusion, etc.), it is unclear how these results will impact the behaviors discussed in the introduction of the manuscript. This absence may limit the broader impacts of this work.

Reviewer #2

(Remarks to the Author)

Reviewer #3

(Remarks to the Author)

This work reports a two-layer FeO interfacial phase, which is called "complexion" by the authors, at the Fe/Fe₃O₄ interface, using 4D-STEM based DPC-STEM imaging. Along with DFT calculations, the FeO interfacial phase is proposed to be highly dependent on oxygen pressure and affects the charge transfer between Fe and Fe₃O₄. Although the manuscript provides valuable atomic structural information at the Fe/Fe₃O₄ interface, it still lacks of a full picture of the interface dynamics. To support their claims, the following issues need to be considered carefully.

1) The authors used 4D-STEM based DPC-STEM to imagine the interface structures, as it is suggested that the potential maps could provide contrast for both O and Fe. However, 4D-STEM images in the current manuscript only provided very localized information, spanning just a few nanometers. The authors should include images with larger fields of view to present a more comprehensive overview of the film. With this regard, traditional ADF-STEM and ABF-STEM might be more suitable. ABF-STEM can also image both O and Fe. Compared to 4D-STEM, ADF/ABF-STEM offers faster scanning rates, which can provide larger fields of view and eliminate scanning distortions in the experiments. Additionally, ABF-STEM imaging could address the issue of low pixel resolution (large pixel sizes) found in the images of the current manuscript.

2) The authors referred to the FeO interfacial phase as "complexion" and devoted considerable spaces in the abstract and introduction to explaining this concept, which makes the paper difficult to read. In fact, the concept of "complexion" is still a subject of debate (e.g., Nature Materials 20 (2021) 907; Nano Letters 21 (2021) 2745-2751).

3) Could the authors use EELS to measure the valence state of Fe at the interface to evidence FeO? Additionally, EDS could detect changes in the O atomic ratio across the interface.

4) In Fig. 2, DFT calculations indicated charge transfer at the Fe-FeO-Fe₃O₄ interface. Could the authors experimentally verify this charge transfer using DPC-STEM?

5) The existence and thickness of FeO phase depended strongly on O₂ pressure. In Fig. 3, the films in panels c and d were fabricated under different O₂ pressures, yet the same 2-layer FeO interfacial phases were formed. How could these results be explained? Additionally, in Fig. 4, a thicker FeO layer was shown, which was attributed to local fluctuations in O₂ activity. If this is the case, the authors should also observe FeO layers of varying thicknesses, such as 3, 4, or 5 layers. Furthermore, the images in Fig. 4 and Supplementary Fig. 14 only showed the FeO-Fe interface; a larger field of view image, showing the Fe₃O₄-FeO-Fe interface, would be more promising.

6) In Fig. 4 ii, the dark field image showed different contrasts between the region closest to Fe (e.g., 2-3 layers) and the region further away from Fe. However, the authors considered all these regions to be FeO. How could one interpret these contrast differences?

Reviewer #4

(Remarks to the Author)

Xuyang Zhou et al. present a combined DPC-4DSTEM and DFT study on an interface-stabilized phase between Fe and Fe₃O₄. The DPC-4DSTEM reconstructions demonstrate the interface-stabilized phase is a FeO-like phase. The DFT calculations show the stability of the interfacial FeO phase depends on the oxygen chemical potential and adjacent bulk phases. This study would be wide of interest since the finds reflect the tunable phase transitions for the initial stage of redox reaction, which is very important for corrosion protection and catalysis. However, there are some critical evidence and improved details that the author needs to provide before any publications can be considered.

1. The orientations of Fe and Fe₃O₄ are directly marked in FIG 1b. The author needs to provide diffraction images for the Fe/Fe₃O₄ interface to verify the orientation relationship between Fe and Fe₃O₄.

2. The interfacial FeO phase is only curtly determined by the DPC images from a single zone axis in FIG 1b. The author at least needs to provide additional DPC images from another zone axis to determine a phase.

3. The O-rich limit is the default $\Delta\mu_{\text{O}} = 0$ and the O-poor limit is not mentioned in FIG 1c. The author needs to consider the limits of $\Delta\mu_{\text{O}}$ for the formations of FeO, Fe₃O₄ and Fe₂O₃ on the Fe surface.

4. The G-type AF magnetic order for the interfacial FeO phase is shown in FIG 1b. G-type AF magnetic order is preferred in the bulk FeO but not always the same in thin films. The magnetic orders in the adjacent bulk Fe and Fe₃O₄ have strong effects on the magnetic order in such a thin film of FeO (1-4 atomic layers). The author needs to verify the preferred magnetic order for the interfacial FeO phase.

Version 1:

Reviewer comments:

Reviewer #1

(Remarks to the Author)

The authors have provided a detailed and comprehensive response to all reviewer comments. We are satisfied with the authors response; the revised manuscript meets our approval for publication and we congratulate the authors on their excellent work.

Reviewer #2

(Remarks to the Author)

Reviewer #3

(Remarks to the Author)

The authors have addressed some concerns in the revised manuscript, in particular the atomic model for the complexion. However, I remain unconvinced that an interfacial FeO phase forms at the Fe/Fe₃O₄ interface based on the DPC-STEM images they provided. Further clarification on this critical issue seems necessary.

First, as depicted in the atomic models (Fig. 1b v-vi and Fig. 4a), the FeO phase should exhibit a zigzag alignment of Fe and O atomic columns along the horizontal direction, where the Fe and O atoms in the same row differ in y-height. In contrast, the experimental DPC-STEM images (e.g., Fig. 1b iv, Fig. 3c-d, and Fig. 4b) show Fe and O atomic columns in the same row at nearly identical heights, and this is inconsistent with the proposed atomic models. Additionally, as evident from the large-field-of-view HAADF images (e.g., Fig. 18 in the supporting materials), the Fe/Fe₃O₄ interface appears highly complex, featuring curvature and inclination. This complexity means that the observed contrast variations at the interface may arise from the overlap of Fe and Fe₃O₄ phases along the thickness direction, rather than the presence of an FeO interfacial phase.

Second, the authors rely exclusively on DPC-4DSTEM images for atomic-resolution analysis, which suffer from poor pixel resolution and limited field of view. Although the authors argue that traditional imaging techniques, such as ABF-STEM, are challenging due to complex contrast transfer functions, the example in Fig. R3, they provided, does not solidly support this claim. The ABF-STEM image in Fig. R3 clearly suffers from sample mis-tilt, which could have been avoided. In fact, ABF-STEM can provide high-quality contrast for both Fe and O in Fe₃O₄-related materials, as demonstrated in the literature (e.g., ACS Nano 2018, 12, 2662–2668; ACS Nano 2021, 15, 19938–19944; Acta Materialia 2024, 271, 119897). Compared to the DPC-4DSTEM images in this paper, ABF-STEM offers superior pixel resolution and a larger field of view, which could better support the authors' claims. While 4D-STEM is indeed a novel and promising technology, but it does not seem well-suited for the objectives of this work.

Finally, in Fig. 4, the authors claim to observe a thick FeO-type interfacial phase but fail to show its precise thickness or details about the FeO/Fe₃O₄ interface. An interfacial phase is typically thin, as it forms only with the support of bulk phases. The reported thick condition in Fig. 4 is more characteristic of a bulk phase than an interfacial one.

Reviewer #4

(Remarks to the Author)

The authors have addressed most of the remarks that I had made except for the images in Figure R6. The images shown for the Fe₃O₄ oriented in the [210] zone axis are not in accord with the cubic phase for Fe₃O₄. I suggest the authors show the DPC images from Fe[110] and Fe₃O₄[100] orientations in the main text. Though the Fe overlaps with O in this orientations, the FeO-like phase can be distinguished from the Fe matrix by comparing the lattice parameter along [001]. The interfacial FeO-like phase can be confirmed clearly after the authors provide the variation of lattice parameter.

Version 2:

Reviewer comments:

Reviewer #3

(Remarks to the Author)

The authors have interpreted their considerations for the major concerns raised in previous reviews, supplemented additional experimental data, reorganized the figures, and reworded descriptions in the main text. Overall, the current version is readable and acceptable for publication.

Reviewer #4

(Remarks to the Author)

My concerns are all addressed, and I think the paper can be accepted.

RE: NCOMMS-24-43961A
Complexions at the Iron-Magnetite Interface

Author response to reviewer comments

REVIEWER COMMENTS

Reviewer #1: The submission titled “Complexions at the Iron-Magnetite Interface” discusses an experimental and computational study on the formation and stability of phase boundary complexions between iron and iron oxide in multilayer films. The key observation was that phase boundary complexions between Fe and Fe oxides in a PVD thin film structurally transitioned as a function of oxygen chemical potential. The results were supported by density functional theory calculations, and the potential impact on bulk material properties was presented. This was a well-written manuscript with exemplary characterization results.

Response: We highly appreciate the thoughtful summary of our work and the recognition of the clarity of our presentation. The positive comments to our experimental and computational analysis are very encouraging. We are grateful for the insights and look forward to revising our manuscript with the valuable feedback.

Detailed comments. Comments 3-6 need to be addressed:

Reviewer #1 Comment 1 & Comment 2: The introduction nicely reviews the importance of bulk oxide materials and their interfaces, and sufficiently introduces the framework of grain boundary complexions to conceptualize interfacial transformations and properties. The necessity of advanced electron microscopy techniques to study complexions in oxide systems was all well formulated. The experimental results are excellent, particularly the use of differential phase contrast (DPC) to image the positions of oxygen-rich columns, as well as extracting projected potentials to most adequately compare to the density functional theory calculations.

Response: We are most grateful for the insightful remarks on our introduction and the acknowledgment of our complementary methodological approach combining DPC and DFT calculations.

Reviewer #1 Comment 3: A major source of confusion in the manuscript is the inconsistent use of terminology. The literature comprises of two nomenclatures that are intended to describe to the same phenomena, namely grain boundary ‘complexions’ and grain boundary ‘phases’. Based upon the title and the rationale discussed by Cantwell et al in their comprehensive review paper on grain boundary complexions we recommend that the authors use the complexion terminology. For example, the section titled ‘The Fe/Fe₃O₄ complexion and its associated interfacial phase diagram’ contains both grain boundary complexions and phases. An option would be to call the thermodynamic diagrams as ‘complexion diagrams’.

Response: We appreciate the valuable suggestions from the reviewer, which indeed help to improve the readability and precision of our manuscript. We acknowledge the inconsistency in our terminology between 'grain boundary complexions' and 'grain boundary phases', which could, indeed, lead to confusion. Based on the comprehensive papers and reviews by Cantwell et al. on grain boundary

complexions (e.g. Cantwell et al., *Acta Materialia*, 62, 1, 2014; Cantwell et al. *Annual Review of Materials Research*, 50, 465, 2020), and the feedback provided, we fully comply and now adopt the 'complexion' terminology consistently throughout our manuscript, as suggested.

Furthermore, we now explicitly clarify in the manuscript that the terms 'complexion' and 'defect phase' have both been used in the literature to describe similar phenomena at grain boundaries. Our choice of 'complexion' aligns with the recent authoritative review and our own experimental observations.

In light of the reviewer's comment, we have now replaced 'interfacial phase diagram' with 'complexion diagram' in the relevant sections. Additionally, we now add a brief statement in the introduction section, explaining the rationale behind our choice of terminology, ensuring that readers are aware of the ongoing debate and the reasons for our specific choice (as highlighted in **Reviewer #3 Comment 2**). This clarification will not only bring coherence to the terms used, but will also improve the scientific accuracy of our manuscript by aligning it with current authoritative perspectives in the field.

Changes made in response to the comment:

'Interfacial phase diagram' has been replaced by 'Complexion diagram' throughout the manuscript.

Reviewer #1 Comment 4: The main issue in our opinion is whether the complexions observed are in equilibrium, and thus the stable complexion type. In the reviewer's opinion, the authors should provide evidence or a strong argument that the 'multilayer of FeO' is a complexion rather than a bulk phase that 'precipitates' on the phase boundary. The results currently only show as-deposited films. Long term heat treatments should be performed to assess the complexion stability and if the structure observed currently is maintained (or different). Another source of confusion is that the authors use FeO to describe the complexion, which is a bulk phase. The authors should not use dashed lines to draw both sides of the complexion. A complexion is the interface itself (one interface) as compared to a wetting film that is comprised of two newly created interfaces (phase boundaries). The DFT work is suspected to artificially reinforce the notion that the observed interfaces are composed of a bulk phase at the interface since FeO serves as a transition 'zone' in valence state of Fe.

Response: We sincerely appreciate the critical and most pertinent insights regarding the question about the equilibrium status of the complexions observed in our study. Our original experimental setup consisted of sputtering at 300 °C for a duration of 2 hours and 15 minutes, which we believe is adequate to approach atomic equilibrium at the interfaces. The diffusion lengths of Fe cations in magnetite and body-centered cubic iron at 300 °C are approximately 20 nm and up to 1 μm depending to the oxygen activity (calculated for 1 minute), respectively (according to F. S. Buffington, K. Hirano and M. Cohen, *Acta Metall.* 9, 434, 1961 for BCC Fe, and R. Dieckmann and H. Schmalzried, *Ber. Bunsen Ges. Phys. Chem.* 81, 344, 1977 for Fe₃O₄, respectively), indicating sufficient mobility for atomic rearrangement at the interface during the deposition process.

To further address the reviewer's concerns about whether the "multilayer of FeO" indeed represents a stable complexion rather than a bulk phase precipitating at the phase boundary, we performed additional in-situ TEM heat treatment experiments to directly monitor any structural changes that might occur at the interface. The experimental conditions included 2 hours of in-situ TEM heating at 300 °C. These

treatments were carefully managed to prevent high temperatures that could induce excessive interdiffusion and potentially alter the local equilibrium conditions.

Our results, as depicted in Fig. R1 (new Supplementary Fig. 18), show both overview and atomic-resolution imaging. Overall, the structure remains unchanged after the prolonged heating; although we observed some local reorganization through a ledge growth mechanism at the atomic scale, the FeO layers consistently appear at the interface. This persistent presence of FeO layers indicates that they are stable features of the interface, reinforcing the picture that they represent an equilibrium complexion rather than a transient phase.

Fig. R1: Comparison of the interface structure between Fe and Fe₃O₄ before and after in-situ heating experiments. **a** Overview HAADF imaging of the Fe-Fe₃O₄-Fe structure in the as-prepared condition. **b** The same structure after in-situ heating in TEM at 300°C for 2 hours, applying heating and cooling rates of 5°C/s, respectively. Both images were taken at room temperature. **c** & **d** Magnified HAADF images that show the atomic structure of the interface between Fe₃O₄-[110] and Fe-[100].

Moreover, subsequently conducted Differential Phase Contrast (DPC) measurements reaffirmed the presence of the FeO-like complexion, consistently showing a distinct three-layer structure at the interface, as illustrated in Fig. R2 (new Supplementary Fig. 19). These findings support our assertion that the interfaces we observed do indeed represent complexions, with FeO serving effectively as a stable transitional zone in the valence state of iron, rather than merely a bulk phase precipitating at the boundary.

Fig. R2: Experimental DPC-4DSTEM reconstruction for a three-layer thick FeO slab at the Fe/Fe₃O₄ interface, with Fe₃O₄ oriented in the [110] direction. **a** Reconstructed virtual dark-field image. Change of the center of mass of the transmitted beam in **b** X and **c** Y directions. **d** Electric field vector map. **e** Projected electrostatic potential map. **f** Charge-density map. The scanning step size used in this experiment was 18 pm. The specimen is the same as shown in Supplementary Fig. 18, which underwent in-situ heating in TEM at 300°C for 2 hours.

The reviewer has correctly noted that we use the structure term ‘FeO’ to describe the complexion, which may cause confusion since FeO is also known as a bulk phase. Although this complexion shares the same chemical formula as the well-known bulk phase FeO, it must be noted that FeO is indeed not a stable bulk phase under our current and heat treatment synthesis conditions, as confirmed by our thermodynamic calculations (see Supplementary Fig. 12). This is actually one of the main novelty items of this work, because the lowest temperature for the bulk existence of this phase is 570°C, i.e. much higher than what we observe here. We show here that such an FeO-like complexion can only be stabilized in the interface regions between the Fe and Fe₃O₄ phases due to the unique local environment and bonding conditions that exist at this very interface. Therefore, we consider it as a complexion rather than a standard bulk phase. A similar phenomenon has been reported in the literature (Baram et al., *Science*, 332, 206, 2011), where nanometer-thick equilibrium films (intergranular films) exist at the gold-sapphire interface. These

intergranular films are not bulk phases but rather interface states stabilized by interfacial energy minimization.

Regarding the reviewer's concern that our DFT work might artificially reinforce the notion that the observed interfaces are composed of a bulk phase at the interface since FeO serves as a transition 'zone' in the valence state of Fe, we would like to clarify that our DFT calculations were specifically designed to model the interfacial region. The FeO-like layer in our DFT models represents a unique interfacial structure with properties distinct from bulk FeO. The calculations show that the FeO-like layers at the interface are energetically favorable and stabilize the interface by accommodating the lattice mismatch and facilitating the transition in the valence state of Fe between metallic Fe and Fe₃O₄. Thus, our DFT results support the existence of a stable interfacial complexion rather than suggesting the presence of a precipitated bulk FeO phase.

We greatly appreciate the very careful and detailed attention of the reviewer to the graphical representation of the complexion. We fully agree that we should not use dashed lines to denote both sides of the complexion, as this can confuse readers by implying the presence of two interfaces. Since a complexion is the interface itself, we have removed the dashed lines and instead used a colored area to emphasize the complexion in our figures.

Once again, we thank the reviewer for the highly valuable feedback, which has greatly helped us to clarify and strengthen our manuscript.

Changes made in response to the comment:

We have updated Figs. 1 – 4 to remove the dashed lines previously drawn on both sides of the complexion and added supplementary Figs. 18 and 19.

Page 16, line 494-504, added: We conducted in-situ heating experiments to examine the stability of the FeO-type complexion. Using the lift-out procedure with an FEI Helios Nanolab 600i FIB dual-beam microscope, we prepared the TEM lamellae on a DENSSolutions chip. We then performed electron microscopy experiments in the Cs probe-corrected FEI Titan Themis 60-300 microscope using similar operating parameters as previously mentioned. The experimental conditions included 2 h of in-situ TEM heating at 300 °C, with a heating and cooling rate of 5 °C/s. Overall, the structure remained unchanged after prolonged heating; although we observed some local reorganization through a ledge growth mechanism at the atomic scale, the FeO-type complexion consistently appeared at the interface (see Supplementary Fig. 18 and Supplementary Fig. 19).

Reviewer #1 Comment 5: The atom probe tomography data is inconsistent with the conclusions that the phase boundary complexion is uniform despite the growing surface. Figure 3 claims that the complexion is identical if the Iron/Oxygen layer grows on the Iron layer and vice versa. However, the atom probe tomography shown in Supplemental Figure 15 shows a shoulder on one interface. Is the atom probe tomography data necessary?

Response: We agree with the reviewer. The use of APT includes significant artifacts due to the different evaporation field of the material adjacent to the interface, creating a distortion effect which is known as local magnification effect. This effect is a characteristic feature of many APT data sets, where the

specimen acts as the actual lens and local evaporation field as well as topologically originated ion flight path alterations can lead to reconstruction artefacts of this type. This is the major reason why we suggested to only place the APT data in the supplementary section. We will follow the suggestions from the reviewer and remove the section relevant to APT measurements from the supplementary information to the revised manuscript.

Changes made in response to the comment:

We have removed the APT-related results from the revised manuscript.

Reviewer #1 Comment 6: Supplemental Figure 3 shows HAADF micrographs of the iron | iron/oxygen interface, but there is no explanation on why these interfaces are non-planar. How does this data compare to the 4D STEM results shown in Figure 1?

Response: We appreciate the reviewer taking the time to read our manuscript so carefully and for raising points that were indeed not fully explained in the text before.

The Fe[001] films were epitaxially grown on MgO[001] substrates. Initially, the layers are coherent and planar. However, as the film thickens, the accumulated strain due to the lattice mismatch of approximately 3.8% can lead to relaxation through non-planar formations of the interfaces. This strain relief can occur via the formation of misfit dislocations, which often manifest as non-planar interfaces in the film, possibly adopting a Stranski-Krastanov growth mode where the initial wetting layer is followed by the development of islands. At the atomic scale, the film can grow with steps and ledges whose growth rates vary locally, leading to inclined interfaces, as reported in Supplementary Figs. 1 and 2.

Supplementary Figure 3 illustrates these non-planar interfaces resulting from strain relaxation. In contrast, Figure 1 presents data from a flat region. We chose such a region for the following reasons:

1. It is locally free from strain and defects, allowing us to isolate influencing factors and focus on how complexions form at the iron/iron-oxide interface.
2. High-resolution imaging and DPC measurements require very accurate edge-on imaging conditions, making planar regions mandatory.
3. A clean model interface enables direct comparison with DFT calculations, providing synergistic atomic insights into the mechanisms behind such interfacial complexions.

Therefore, while Supplementary Figure 3 shows the non-planar interfaces due to strain relaxation, Figure 1 focuses on a planar region to study the formation and structure of interfacial complexions under controlled conditions.

Changes made in response to the comment:

Page 14, line 429-434, added: In this work, we specifically study the defect-free and strain-free interface to isolate factors influencing complexion formation. This clean interface structure (see Fig. 1 and Fig. 3) not only facilitates accurate high-resolution imaging and DPC measurements but also enables direct comparisons with DFT calculations to provide deep insights into the atomic mechanisms at play.

Reviewer #1 Comment 7: Finally, while bulk material properties were discussed in the introduction, there is a lack of physical measurements in this work. Without measurements of property data (mechanical properties, diffusion, etc.), it is unclear how these results will impact the behaviors discussed in the introduction of the manuscript. This absence may limit the broader impacts of this work.

Response: We appreciate the reviewer's comments regarding the absence of physical measurements. While the current manuscript primarily targets and investigates the basic phenomena in its very nature, we have nonetheless provided also some theoretical predictions to guide future experimental efforts along these lines. Metal/oxide interfaces have broad implications for many practical fields (e.g. corrosion, green ironmaking, catalysis, magnetism, etc.), and perhaps further emphasis on a specific physical measurement could detract from the purpose of understanding the interfacial structure broadly. We do have ongoing research focused on exploring diffusion processes and their impact on the reduction of green steel. This upcoming work aims to directly address the practical implications of our findings and contribute further to the field of materials sustainability.

Therefore, we chose - in the present manuscript - to present primarily the basic discovery of this phenomenon in such Fe-oxides, its structural characteristics, the theoretical rationale behind it and the atomistic origin of these FeO-like complexions, which, as they change the structure of the Fe/Fe₃O₄ interface, will for sure have impact on some of the reduction properties, for instance, such as mentioned by the reviewer.

Reviewer #2: I co-reviewed this manuscript with one of the reviewers who provided the listed reports. This is part of the Nature Communications initiative to facilitate training in peer review and to provide appropriate recognition for Early Career Researchers who co-review manuscripts.

Response: We highly appreciate the reviewer for co-reviewing our manuscript. We will carefully address the comments provided in the listed reports.

Reviewer #3: This work reports a two-layer FeO interfacial phase, which is called "complexion" by the authors, at the Fe/Fe₃O₄ interface, using 4D-STEM based DPC-STEM imaging. Along with DFT calculations, the FeO interfacial phase is proposed to be highly dependent on oxygen pressure and affects the charge transfer between Fe and Fe₃O₄. Although the manuscript provides valuable atomic structural information at the Fe/Fe₃O₄ interface, it still lacks of a full picture of the interface dynamics. To support their claims, the following issues need to be considered carefully.

Response: We are grateful to the reviewer for the thoughtful review and concise summary of our work. Our study focuses on understanding the FeO-type complexions at the Fe/Fe₃O₄ interface and explaining their formation at phase boundaries. We acknowledge the importance of the possibly associated interface

dynamics and have therefore fully complied with the reviewer and conducted additional in-situ heating experiments directly in the TEM, with down to atomic scale probing, to explore this aspect further. Please refer to our detailed response to **Reviewer #1 Comment 4** for more information. The current work lays the groundwork for future studies in areas such as oxide reduction, initial corrosion, and catalysis. Further detailed investigation of these topics is beyond the scope of this manuscript. We will address the remaining comments point by point in our subsequent responses.

Reviewer #3 Comment 1: The authors used 4D-STEM based DPC-STEM to image the interface structures, as it is suggested that the potential maps could provide contrast for both O and Fe. However, 4D-STEM images in the current manuscript only provided very localized information, spanning just a few nanometers. The authors should include images with larger fields of view to present a more comprehensive overview of the film. With this regard, traditional ADF-STEM and ABF-STEM might be more suitable. ABF-STEM can also image both O and Fe. Compared to 4D-STEM, ADF/ABF-STEM offers faster scanning rates, which can provide larger fields of view and eliminate scanning distortions in the experiments. Additionally, ABF-STEM imaging could address the issue of low pixel resolution (large pixel sizes) found in the images of the current manuscript.

Response: We appreciate the insightful suggestions from the reviewer regarding the need for images with larger fields of view to present a more comprehensive overview of the film. We have indeed provided such overviews in Supplementary Fig. 1, which includes an illustrative representation of the multilayer thin film, high-angle ADF-STEM images, and EDS scans. Supplementary Fig. 2 presents the phase and orientation maps of the multilayer thin film. We hope these images address the concerns of the reviewer and provide a broader perspective of the sample. If the reviewer refers to a larger field of view with the atomic resolution, we want to further guide the reviewer to our response to **Reviewer #1 Comment 4**. These images not only show the interface structure, but also provides snapshots capturing the dynamic change of the interface structure at 300 °C for 2 hours.

Regarding the suggestion to use ABF-STEM imaging, we agree that ABF-STEM can provide contrast for both oxygen and iron atoms (J. Fertig and H. Rose, *Optik* 54, 1979; W. Saxton, et al., *Optik (Jena)* 49, 505, 1978; E. Okunishi, et al., *Microscopy and Microanalysis* 15, 164, 2009). However, ABF-STEM measurements of phase contrast are challenging due to their complex contrast transfer functions (C. Ophus, *Annual Review of Materials Research* 53, 105, 2023). We encountered significant challenges in achieving high-resolution images of the Fe–O system using ABF-STEM on the Cs probe-corrected FEI Titan Themis 60-300 microscope. Specifically, the ABF-STEM imaging was highly sensitive to the alignment between the bright-field disc and the dark-field detector (DF2). Even slight misalignments led to substantial variations in image contrast, making it difficult to obtain consistent and high-quality images. Despite experimenting with various semi-collection angles (13-21, 10-16, 8-13 mrad), we were unable to achieve satisfactory contrast in the magnetite regions, and even less so in the interface regions of interest. Fig. R3 shows an example of the ABF-STEM image we obtained for the [011]-oriented

specimen, illustrating these limitations (semi-collection angles: 13-21 mrad).

Fig. R3: Dark-field imaging of Fe_3O_4 oriented in the $[110]$ zone axis. a. HAADF image, b. reverse-annular bright field (r-ABF) image, and c. the corresponding illustration of the atomic structure.

Therefore, we opted to use differential phase contrast (DPC) STEM imaging as the primary tool for visualizing the light oxygen atoms. DPC-STEM allowed us to achieve spatial resolution and contrast when imaging the Fe–O interfaces.

Regarding the issue of low pixel resolution and large pixel sizes in our images, we acknowledge this limitation. The high-resolution 4D-STEM datasets required longer acquisition times, which constrained the field of view to a few nanometers to minimize beam damage and sample drift. We aimed to balance the resolution and field of view while preserving the integrity of the sample.

We hope that our explanations adequately address the concerns of the reviewer.

Changes made in response to the comment:

Page 15, line 468-473 added: Annular bright field (ABF)-STEM imaging concurrently provides contrast for both heavy Fe atomic columns and light O atomic columns [73, 74]. However, ABF-STEM phase contrast measurements are challenging due to complex contrast transfer functions [75]. In this study, we consistently employed DPC-4STEM as the primary method for imaging the interface structure.

Reviewer #3 Comment 2: The authors referred to the FeO interfacial phase as "complexion" and devoted considerable spaces in the abstract and introduction to explaining this concept, which makes the paper difficult to read. In fact, the concept of "complexion" is still a subject of debate (e.g., Nature Materials 20 (2021) 907; Nano Letters 21 (2021) 2745-2751).

Response: We sincerely thank the reviewer for the insightful comments regarding the use of the term "complexion" in our manuscript. We acknowledge that the concept of the "complexions" is a subject of ongoing debate in the materials science community, as highlighted by the references you provided (Nature Materials 20, 907, 2021; Schusteritsch et al., Nano Letters 21, 2745, 2021). We have cited these

references in our revised manuscript to adequately reflect the nature and state of the art of this debate and also to provide the full context and acknowledge the different perspectives on this topic.

Our experimental observations and atomistic simulations support the existence of FeO-like interfacial complexions at the Fe/Fe₃O₄ interface. We believe that using the term "complexion" accurately describes the interfacial phenomena we have observed here and aligns with recent studies in the field, as also acknowledged and supported by the other reviewers. To improve the readability of the paper, we have streamlined the abstract and introduction by making the explanation of the "complexion" concept more concise while still providing sufficient background for readers.

Additionally, based on feedback from other reviewers who suggested consistent use of the "complexion" terminology (for example, **Reviewer #1** recommended that "the authors use the complexion terminology"), we have decided to retain the term throughout the manuscript. We believe that maintaining consistent terminology and explaining what it refers to in the context of our manuscript help clarify our findings and contributes constructively to the ongoing discussion in the field.

We hope that these revisions address your concerns and improve the overall readability of the paper. We are grateful for the thoughtful feedback.

Changes made in response to the comment:

Page 3, Line 79-83, added: Although the term "complexion" has gained acceptance for a range of interfacially-stabilized phenomena [29, 33–36], its terminology still invites debate [37, 38]. In this work, we consistently use the word "complexion" aligning with our experimental observations and recent reviews [30, 31] to describe these interfacial phenomena accurately.

Reviewer #3 Comment 3: Could the authors use EELS to measure the valence state of Fe at the interface to evidence FeO? Additionally, EDS could detect changes in the O atomic ratio across the interface.

Response: We appreciate the thoughtful suggestions from the reviewer regarding the use of EELS to measure the valence state of iron at the interface, and EDS to detect changes in the oxygen atomic ratio across the interface. These are indeed valuable techniques for improving the characterization of the interface.

Following the reviewer's suggestion, we have now conducted additional EELS analysis to determine the valence state of the iron. The evaluation of the Fe³⁺ ions to the total iron ratio was guided by the methodology outlined in the reference 'van Aken et al., *Physics and Chemistry of Minerals* 29, 188, 2002'. For the valence calculation, we conducted peak decomposition of the Fe L₃ edges to quantify the fraction of different oxidation states.

As illustrated in Fig. R4 (new Supplementary Fig. 20), the EELS analysis, along with the HAADF imaging, provides insights into the O atomic ratio and the valence state transitions across the interface. Notably, Fe³⁺ is not entirely absent, which likely indicates the formation of surface oxides, predominantly Fe₃O₄, as Fe tends to oxidize readily. Although the complexion region is discernible in HAADF images, we observed a gradient rather than a sharp transition in the valence states at and near the interface. This gradual transition is primarily due to beam broadening effects, and the overlapping signals of Fe, Fe²⁺, and Fe³⁺ within the Fe₃O₄ inverse spinel, a circumstance which complicates the direct observation of

complexions using EELS. Despite multiple attempts, this was the clearest resolution we could achieve under our current experimental conditions. A higher base TEM, such as a Titan with superior EELS capabilities, might yield better resolution, though such an instrument is not currently available at our institute.

Regarding quantitative measurements related to light elements, we find EELS to be more reliable than EDS, particularly for distinguishing subtle changes in elemental states and ratios. This preference is based on EELS's enhanced sensitivity to light elements and its ability to provide detailed electronic structure information, which is crucial for accurately identifying different valence states at the interface.

We trust that these additional data and explanations will satisfactorily address the reviewer's query and substantiate the interpretations made in our study.

Fig. R4. Local chemistry and charge state analysis for complexions at the Fe-Fe₃O₄ interface. a.

Dark-field image serves for electron energy loss spectroscopy (EELS) analysis of the Fe-Fe₃O₄ interface.

b. Selected energy loss spectra for scanning regions across the Fe-Fe₃O₄ interface correspond to regions in the dark-field image from a. The spectra highlight the O-K edge and Fe-L edges. **c.** Quantified local composition using EELS for regions depicted in a, with distances from 0-4.2 nm matching the dark-field

image from top to bottom. **d.** Magnified regions of the spectra from **b** show the Fe-L₃ peak, revealing a shift to a higher energy state from the Fe region (upper part) to the Fe₃O₄ (lower part) region. **e.** Quantified the charge state using EELS for the regions shown in **a**. **f** and **g** display selected spectra for peak decomposition in the Fe region and the Fe₃O₄ region, respectively, using three Gaussian peaks (pink, purple, and yellow) for decomposition. The ratio of Fe³⁺/ΣFe inversely relates to the integrated area of the pink peak [79]. **h-j** present the dark-field image for the EELS analysis and quantifications of local composition and charge state for the scanning regions across the Fe₃O₄-Fe interface.

Changes made in response to the comment:

We have added Supplementary Fig. 20 for showing, quantifying and discussing our EELS measurements.

Page16, Line 505-517, add: For quantifying local composition and valence state transitions across the interface, we used a Gatan Quantum spectrometer to acquire electron energy loss spectroscopy (EELS) data at 300 kV with an entrance aperture of 35 mrad. We estimated the atomic ratio of Fe to O by integrating the intensities of the Fe-L₃ and O-K edges. Additionally, we performed peak decomposition on the Fe-L₃ edges to determine the fractions of various oxidation states [79]. As Supplementary Fig. 20 illustrates, EELS analysis provides insights into the O atomic ratio and valence state transitions across the interface of the specimen heated in-situ at 300 °C for 2 h. Although the complexion region appears in HAADF images, a gradient in valence states at the interface, rather than a sharp transition, occurs due to beam broadening and overlapping signals of Fe, Fe²⁺, and Fe³⁺ in Fe₃O₄, complicating the direct observation of complexions using EELS.

Reviewer #3 Comment 4: In Fig. 2, DFT calculations indicated charge transfer at the Fe-FeO-Fe₃O₄ interface. Could the authors experimentally verify this charge transfer using DPC-STEM?

Response: We appreciate the reviewer's insightful question regarding the experimental verification of charge transfer at the Fe-FeO-Fe₃O₄ interface using DPC-STEM. This is indeed an intriguing aspect that we considered during our research.

DPC-STEM is a powerful technique for mapping electric and magnetic fields at the nanoscale by detecting deflections of the electron beam caused by local electromagnetic fields within the specimen. However, using DPC-STEM to directly measure charge transfer at atomic interfaces, such as between iron and its oxides, presents significant challenges.

The primary reason is that DPC-STEM is sensitive to the total electrostatic potential, which is dominated by the cumulative charge density of all electrons in the atoms—including both core and valence electrons. In the case of iron, each atom has 26 electrons, and oxygen has 8 electrons. The loss or gain of a few valence electrons (e.g., 2–3 electrons involved in charge transfer) constitutes a very small change relative to the total electron count. This minute variation is further overshadowed by the strong electrostatic potential from the atomic nuclei and inner-shell electrons, making it extremely difficult to detect subtle charge redistributions using DPC-STEM.

Moreover, the spatial resolution and sensitivity required to detect such small changes in charge density at the atomic scale exceed the current capabilities of DPC-STEM. The technique is more suited for

observing larger-scale electric and magnetic fields rather than the fine-scale charge transfer between specific atoms in a crystal lattice.

Therefore, while DPC-STEM is effective for imaging light elements and mapping electromagnetic fields, it is not feasible to experimentally verify the specific charge transfer at the Fe–FeO–Fe₃O₄ interface using this method.

We hope this explanation clarifies the limitations of the method, and we appreciate the reviewer's understanding of the technical challenges involved.

Reviewer #3 Comment 5: The existence and thickness of FeO phase depended strongly on O₂ pressure. In Fig. 3, the films in panels c and d were fabricated under different O₂ pressures, yet the same 2-layer FeO interfacial phases were formed. How could these results be explained? Additionally, in Fig. 4, a thicker FeO layer was shown, which was attributed to local fluctuations in O₂ activity. If this is the case, the authors should also observe FeO layers of varying thicknesses, such as 3, 4, or 5 layers. Furthermore, the images in Fig. 4 and Supplementary Fig. 14 only showed the FeO-Fe interface; a larger field of view image, showing the Fe₃O₄-FeO-Fe interface, would be more promising.

Response: We sincerely thank the reviewer for the insightful comments and the opportunity to clarify our findings.

The observation that the same 2-layer FeO-like interfacial phase forms under different O₂ pressure pathways is indeed an important aspect of our work. In Figures 3c and 3d, both films were subjected to the same low and high O₂ pressures. The key difference lies in the sequence of the pressure change: one sample experienced a transition from low to high O₂ pressure, while the other underwent a change from high to low. Remarkably, the 2-layer FeO-like phase formed in both cases, indicating that its formation is independent of the O₂ pressure pathway. This phenomenon suggests that the formation of the FeO-like interfacial phase between Fe and Fe₃O₄ is thermodynamically driven rather than kinetically controlled. Our DFT calculations, which are inherently pathway-independent, support this conclusion by predicting the thermodynamic stability of the FeO-like complexion at the interface.

We have indeed observed FeO layers of varying thicknesses, including cases with different number of layers. To address the concern of the reviewer, we have included an additional image in the revised manuscript (new Supplementary Fig. 19), showcasing a 3-layer FeO-like complexion observed after annealing at 300°C for 2 hours in the TEM. This additional data demonstrates that variations in FeO layer thickness do occur and are consistent with our explanation of local O₂ activity fluctuations. Using the new in situ TEM results, we further proved that the formation of interface complexion is not due to phase nucleation and growth but rather to a defect structure at the interface.

We understand the importance of providing a comprehensive image of the Fe₃O₄-FeO-Fe interface. However, capturing a larger field of view while maintaining atomic resolution presents technical challenges. In our previous study (X. Zhou, et al., Nature communications 14, 3535, 2023), we optimized our scanning parameters with an ideal spacing of 18 pm to achieve high resolution while minimizing drift during imaging. In Fig. 4 of the original manuscript, we presented an extreme case to illustrate the potential for thicker FeO-like complexions due to local fluctuations in O₂ activity. Although the Fe₃O₄

layer is not present in this specific figure, it was observed in our experiments. Expanding the field of view under these conditions would compromise image quality due to increased drift and reduced resolution. Additionally, to minimize potential electron beam damage during DPC scanning, we have limited our scans to a single pass over each area. Repeated scans or extended scanning areas increase the risk of sample degradation, which can obscure or alter the features of interest. Therefore, the Fe₃O₄ layer was not included in the original image.

We appreciate the opportunity to clarify these points and believe that the revisions and additional data provided in our manuscript now comprehensively address the concerns raised. We hope that our responses and the updated manuscript will satisfy the queries of the reviewer.

Reviewer #3 Comment 6: In Fig. 4 ii, the dark field image showed different contrasts between the region closest to Fe (e.g., 2-3 layers) and the region further away from Fe. However, the authors considered all these regions to be FeO. How could one interpret these contrast differences?

Response: We appreciate the reviewer for pointing out the observations regarding the contrast differences in the dark field image of Fig. 4 ii. The variations in contrast within the FeO region can be attributed to interface steps and strain contrast, as seen near the middle right side of Fig. 4 ii. Interface steps can cause variations in thickness or density of the FeO layer, thereby affecting the dark field image contrast. Additionally, the virtual collecting angle used for image reconstruction, although high at 101-125 mrad, can still capture crystallographic information such as strain due to lattice mismatches. There is a significant mismatch between Fe and FeO, with $\varepsilon_{xy}^{\text{BCC-Fe}} = -6.8\%$ and $\varepsilon_{xy}^{\text{FeO}} = +7.3\%$ with respect to the other layer. These factors can lead to contrast variations, particularly noticeable near the Fe interface where strain and interface effects are often more pronounced.

Reviewer #4: Xuyang Zhou et al. present a combined DPC-4DSTEM and DFT study on an interface-stabilized phase between Fe and Fe₃O₄. The DPC-4DSTEM reconstructions demonstrate the interface-stabilized phase is a FeO-like phase. The DFT calculations show the stability of the interfacial FeO phase depends on the oxygen chemical potential and adjacent bulk phases. This study would be wide of interest since the finds reflect the tunable phase transitions for the initial stage of redox reaction, which is very important for corrosion protection and catalysis. However, there are some critical evidence and improved details that the author needs to provide before any publications can be considered.

Response: We sincerely appreciate the insightful comments from the reviewer and the acknowledgment of our study's relevance to tunable phase transitions important for corrosion protection and catalysis. We are encouraged by the recognition of the potential broad interest in our findings. In our forthcoming response, we hope to provide such critical evidence and detailed information as requested by the reviewer to fully substantiate our conclusions.

Reviewer #4 Comment 1: The orientations of Fe and Fe₃O₄ are directly marked in FIG 1b. The author needs to provide diffraction images for the Fe/Fe₃O₄ interface to verify the orientation relationship between Fe and Fe₃O₄.

Response: We thank the reviewer for the suggestion, which indeed helps to clarify this aspect to readers. To address the requirement from the reviewer, we have conducted 4D-STEM phase and orientation mapping, detailed in Supplementary Figure 2 of the original manuscript (Fig. R5 in the response letter below). Additionally, we have included three diffraction patterns from the 4D-STEM scan in the revised manuscript: one from the Fe region, another from the Fe₃O₄ region, and a third showing a region mixed with both. These diffraction patterns clearly demonstrate the orientation relationship, with Fe[001]/Fe₃O₄[001] and Fe[100]/Fe₃O₄[110]. This information directly supports the orientation relationship between Fe and Fe₃O₄ as described.

Changes made in response to the comment:

The following Fig. R5 is now provided as Supplementary Figure 2 of the revised manuscript.

Fig. R5: Structural characterization of the multi-layer thin film. **a** Phase map and orientation maps obtained from **b** View A1 and **c** View A3, reconstructed from the precession-assisted 4DSTEM datasets of the cross-sectional view of the thin film. The sample coordinates A1-A3 were defined in the phase map. **d-f** Diffraction patterns from regions highlighted in **c** (orange, blue, and purple circles) are **d** magnetite-[100], **e** the interface between magnetite-[100] and Fe-[110], and **f** Fe-[110]. Additional diffraction spots in **d** and **f**, not part of the Fe or Fe₃O₄ lattice, arise from the native surface oxide layer on Fe.

Reviewer #4 Comment 2: The interfacial FeO phase is only curtly determined by the DPC images from a single zone axis in FIG 1b. The author at least needs to provide additional DPC images from another zone axis to determine a phase.

Response: We appreciate the suggestion of the reviewer to provide additional DPC images from another zone axis to more comprehensively determine the interfacial FeO phase. In the original submission, we have provided DPC measurements from the Fe[110] and Fe₃O₄[100] orientations in Supplementary Figs. 16 and 17. Unfortunately, in this direction, Fe overlaps with O, which compromises the clarity of the complex structure.

The optimal directions for imaging, which we employed primarily in our study, are the Fe[100] and the Fe₃O₄[110] directions, respectively. In this set of orientations, Fe and O atoms are aligned in separate columns, providing hence the clearest visualization of the complex interface structure. These directions not only allow for the best structural resolution, but also offer greater tolerance for experimental conditions, unlike other possible orientations such as Fe[210] and Fe₃O₄[120]. Along these latter orientations, the spacing between closely packed atoms is very small, making it extremely challenging to maintain atomic resolution at the interface, as shown in Fig. R6. Even the slightest astigmatism can result in a loss of atomic detail, making it nearly impossible to keep both phases in the zone axis simultaneously.

Fig. R6: Dark-field imaging of Fe and Fe₃O₄ oriented in the [210] zone axis. a-c for Fe and e-g for Fe₃O₄. Here, a and e are HAADF images, b and f are reverse-annular bright field (r-ABF) images, and c and g are the corresponding illustrations of the atomic structure.

Given these constraints, we focused on the Fe[100] and Fe₃O₄[110] orientation as it allows for a more reliable observation of the interfacial structure, despite the experimental challenges it still presents. We

believe this approach provides the most feasible and effective method to elucidate the complex interface under study.

Reviewer #4 Comment 3: The O-rich limit is the default $\Delta\mu\text{O} = 0$ and the O-poor limit is not mentioned in FIG 1c. The author needs to consider the limits of $\Delta\mu\text{O}$ for the formations of FeO, Fe₃O₄ and Fe₂O₃ on the Fe surface.

Response: As pointed out by the reviewer, the O-rich limit is indeed set to the chemical potential of oxygen in its O₂ molecular form. Usually, the upper limit in the allowed oxygen chemical potential range is set to the decomposition of Fe₃O₄ into metallic Fe and molecular O₂, which we now added as a vertical black dashed line in the complexion diagram of Figure 1c in the revised version of the manuscript. We hope that with this addition, the information supported by the complexion diagram makes more sense in light of the reviewer's comment.

Changes made in response to the comment:

A vertical black dashed line now appears on the revised version of Fig. 1c of the main text, indicating the oxygen chemical potential at which Fe₃O₄ decomposes into metallic Fe and molecular O₂. The legend of the figure has also been updated accordingly.

Reviewer #4 Comment 4: The G-type AF magnetic order for the interfacial FeO phase is shown in FIG 1b. G-type AF magnetic order is preferred in the bulk FeO but not always the same in thin films. The magnetic orders in the adjacent bulk Fe and Fe₃O₄ have strong effects on the magnetic order in such a thin film of FeO (1-4 atomic layers). The author needs to verify the preferred magnetic order for the interfacial FeO phase.

Response: As pointed out by the reviewer, the magnetic order and structure of materials in atomic-sharp thin films might indeed differ from their bulk counterparts. For the original submission, we only checked whether the interface is more stable comparing the case of G-type AF and FM magnetic orders for the FeO-like complexion at the Fe/Fe₃O₄ interface. We found that a G-type AF FeO complexion was indeed more stable than a FM layer by 12 meV/A². Since the G-type AF order is also the ground-state magnetic order of bulk FeO, we thus decided to present results considering only this order of the FeO-like complexions at the Fe/Fe₃O₄ interface.

To further address the reviewer's concern, we have now complemented this stability analysis considering more complex magnetic structures, which are shown in Fig. R7 below. These structures were selected by screening through different possible non-equivalent magnetic orders, and correspond to a lower energy structure than the FM case. We denote this new magnetic order as AF₂ in the following. For this study, we only considered the 1-layer thick FeO-like complexion since we believe it is for the thinnest complexions that such a magnetic effect should be the most pronounced.

Fig. R7: Different magnetic orders of the interfacial FeO-like complex at the Fe/Fe₃O₄ interface. Considering a 1-layer thick FeO-like complexions, showing different magnetic orders of the FeO at the interface: **a** the original bulk G-type AF order of the FeO layer, and **b** the AF₂ magnetic order.

For the 1-layer complexion, the AF₂ magnetic order shown in Fig. R7b above is also antiferromagnetic within the FeO layer, but with a different coupling to the two surrounding Fe and Fe₃O₄ layers. For the FeO-like layer to adopt such a magnetic order, the interfacial energy drops by 11 meV/Å², thus more stable than the original G-type AF order, but rather small compared to the total interfacial energy. At the temperatures at which the experiments were carried out, such a small energy difference between the two antiferromagnetic orders of the FeO layer will be smoothed out by thermal effects.

We also believe, in the context of the experimental observations on which the DFT calculations are based, that since the temperature of both the analysis of the structure and the deposition of the sample is well above the Néel temperature of FeO (approximately 200 K, see for instance McCammon, *J. Mag. Mag. Mat.* 104, 1937, 1992), the FeO-layer might very well already be in a completely disordered paramagnetic state. Thus, the relative stability between different magnetic orders is not expected to play a key role in the driving force responsible for the formation of such FeO-like complexions.

We hope these additional considerations have addressed the reviewer's concern, and have also proven that the magnetic order the FeO-like complexions adopts does not change the picture in terms of stability of such interfacial structures but only play a second order role on the stabilization of the observed complexions.

RE: NCOMMS-24-43961B
Complexions at the Iron-Magnetite Interface

Author response to reviewer comments

REVIEWER COMMENTS

Reviewer #1: The authors have provided a detailed and comprehensive response to all reviewer comments. We are satisfied with the authors response; the revised manuscript meets our approval for publication and we congratulate the authors on their excellent work.

Response: We are most grateful for the reviewer's kind appreciation, approval and positive feedback on our revised manuscript. The previous thoughtful comments and guidance provided have been essential in enhancing our work. We are honored by the congratulations and look forward to our research contributing to the field. We extend our appreciation once again to the reviewer for this esteemed opportunity.

Reviewer #2: I co-reviewed this manuscript with one of the reviewers who provided the listed reports. This is part of the Nature Communications initiative to facilitate training in peer review and to provide appropriate recognition for Early Career Researchers who co-review manuscripts.

Response: We appreciate the kind efforts of the co-reviewer of our paper and value the collaborative effort. We are highly grateful for the opportunity to improve our manuscript through this process.

Reviewer #3: The authors have addressed some concerns in the revised manuscript, in particular the atomic model for the complexion. However, I remain unconvinced that an interfacial FeO phase forms at the Fe/Fe₃O₄ interface based on the DPC-STEM images they provided. Further clarification on this critical issue seems necessary.

Response: We thank the reviewer for acknowledging the improvements in the atomic model of the complexion. We appreciate the concerns raised regarding the formation of an interfacial FeO phase at the Fe/Fe₃O₄ interface based on the DPC-STEM images. We will provide further clarification to address these concerns comprehensively as follows.

Reviewer #3 Comment 1: First, as depicted in the atomic models (Fig. 1b v-vi and Fig. 4a), the FeO phase should exhibit a zigzag alignment of Fe and O atomic columns along the horizontal direction, where the Fe and O atoms in the same row differ in y-height. In contrast, the experimental DPC-STEM images (e.g., Fig. 1b iv, Fig. 3c-d, and Fig. 4b) show Fe and O atomic columns in the same row at nearly identical heights, and this is inconsistent with the proposed atomic models. Additionally, as evident from the large-field-of-view HAADF images (e.g., Fig. 18 in the supporting materials), the Fe/Fe₃O₄ interface appears highly complex, featuring curvature and inclination. This complexity means that the observed contrast variations at the interface may arise from the overlap of Fe and Fe₃O₄ phases along the thickness direction, rather than the presence of an FeO interfacial phase.

Response: We acknowledge the reviewer’s kind concerns about local strain at the interface between Fe and Fe₃O₄, which might indeed result in local distortion within the FeO-like complex. Additionally, during the revision process, and as also detailed in our response to previous remarks from the reviewers, we now consider different magnetic orders for the interfacial FeO-like complexes of various thicknesses. Indeed, after further DFT calculations, we realized that keeping the bulk FeO antiferromagnetic (AF) order was less favorable than considering more complex magnetic ordering states, given the small thickness of the observed FeO-like complexes. In particular, we report a new magnetic order for the two-layer case, which has now been compared to the original bulk FeO AF order in Fig. RR1.

Fig. RR1: Two different magnetic ordering states of an interfacial two-layer FeO-like complex predicted by DFT calculations and compared to experimental DPC charge density maps. a Comparison between the experimental DPC-reconstructed charge density map and DFT-predicted charge density for the original magnetic order of the two-layer FeO-like complex (i.e. considering the bulk antiferromagnetic magnetic order of FeO). **b** Similar to **a**, but considering a different magnetic order for the interfacial two-layer FeO-like complex, having a lower energy than the original one.

This new magnetic order probed here corresponds to a first ferromagnetic (FM) layer on the BCC Fe side of the interface, while the FeO layer on the Fe₃O₄ side remains AF (see Fig. RR1.b). Considering this new magnetic order of the FeO-like 2-layer complex decreases the interface energy by 26 meV/Å² compared to the original bulk AF order considered in the original version of the manuscript. With a different coupling between the layers of the complex, the zigzag distortion is a little less pronounced in the layer close to the Fe₃O₄ side of the interface, more in line with our experimental observations and the reviewer’s comment.

We sincerely appreciate the reviewer’s continued interest and most thoughtful feedback on curvature and inclination. As we noted previously, “It is locally free from strain and defects, allowing us to isolate influencing factors and focus on how complexions form at the iron/iron-oxide interface” and “A clean model interface enables direct comparison with DFT calculations, providing synergistic atomic insights into the mechanisms behind such interfacial complexions.” and “In this work, we specifically study the defect-free and strain-free interface to isolate factors influencing complexion formation. This clean interface structure (see Fig. 1 and Fig. 3) not only facilitates accurate high-resolution imaging and DPC measurements but

also enables direct comparisons with DFT calculations to provide deep insights into the atomic mechanisms at play.” By conducting our detailed analyses in these planar and defect-free regions, we can more confidently attribute the observed contrast variations to actual interfacial features rather than to projection artifacts arising from interfacial curvature or overlapping phases. We acknowledge the complexities that can arise in other regions of the sample. However, by combining a carefully chosen, well-characterized interface location with state-of-the-art imaging and computational approaches, we have taken steps to ensure that the results more accurately reflect the genuine presence and structure of the FeO interfacial complexion.

Changes made in response to the comment:

We have updated Fig. 1 to replace the DFT-calculated charge density map from the original antiferromagnetic ordering with the results that reflect the study conducted on the new magnetic ordering state of the complexions.

Page 17, line 548-557, added: For the BCC-Fe and Fe₃O₄ layers, all magnetic moments are initialized as in the ground-state magnetic order of each structure, namely ferromagnetic (FM) for BCC-Fe and ferrimagnetic (FeM) for Fe₃O₄. As for the FeO-like complexions, we search – by using DFT - for the lowest energy magnetic ordering state by considering different arrangement of the Fe magnetic moments within the complexion layer. We did so since it might well be that the bulk AF order of FeO is not the most energetically favorable, given the thinness of the complexions. Only the lowest energy configurations are presented, and all values derived from these structures, which all differ from the bulk AF magnetic order of FeO.

Reviewer #3 Comment 2: Second, the authors rely exclusively on DPC-4DSTEM images for atomic-resolution analysis, which suffer from poor pixel resolution and limited field of view. Although the authors argue that traditional imaging techniques, such as ABF-STEM, are challenging due to complex contrast transfer functions, the example in Fig. R3, they provided, does not solidly support this claim. The ABF-STEM image in Fig. R3 clearly suffers from sample mis-tilt, which could have been avoided. In fact, ABF-STEM can provide high-quality contrast for both Fe and O in Fe₃O₄-related materials, as demonstrated in the literature (e.g., ACS Nano 2018, 12, 2662–2668; ACS Nano 2021, 15, 19938–19944; Acta Materialia 2024, 271, 119897). Compared to the DPC-4DSTEM images in this paper, ABF-STEM offers superior pixel resolution and a larger field of view, which could better support the authors’ claims. While 4D-STEM is indeed a novel and promising technology, but it does not seem well-suited for the objectives of this work.

Response: We thank the reviewer for the continued feedback and suggestions. The spatial resolution of our HAADF-STEM setup at 300 kV is slightly lower than 80 pm, and for our DPC scan, we use a pixel size of about 18 pm. We believe that this choice achieves the optimal spatial resolution for our instrument. Although we could further reduce the scanning step size, it would not improve the actual pixel resolution. As indicated in the manuscript (see Supplementary Fig. 23), we performed a systematic imaging simulation to determine this scanning pixel size. While increasing the pixel size could broaden the field of view, it would compromise the critical imaging conditions needed for our objectives.

We are grateful for the literature references mentioned by the reviewer and appreciate the reviewer’s suggestion regarding ABF-STEM imaging. As shown in our previous response letter, we attempted to

provide an ABF-STEM image. The reviewer notes that the ABF-STEM image in Fig. R3 suffers from sample mis-tilt. We attribute this issue primarily to residual astigmatism, which prevented us from achieving perfect correction for both HAADF and ABF imaging simultaneously. We needed to balance the correction settings to optimize both HAADF and ABF conditions. Slight deviations from the ideal coaxial alignment, particularly between the HAADF and DF detectors in our TEM setup, further complicated our efforts to achieve optimal ABF imaging while maintaining high-quality HAADF imaging. These practical constraints made it significantly more challenging to achieve the literature-level ABF quality with our setup. While it is possible to obtain outstanding ABF-STEM images under ideal conditions (as demonstrated in the cited literature), doing so with our particular TEM configuration and at interfaces is extraordinarily difficult.

4D-STEM offers several advantages for our specific study at hand here. With a single scan, we obtain comprehensive diffraction data that enable us to reconstruct various imaging modalities (HAADF, BF, DF, ABF, and DPC) from the same dataset. This approach provides richer information than a single imaging method can deliver. We provide reconstructed HAADF, reverse-BF, and reverse-ABF images, along with a reconstructed charge density map, all derived from the same 4D-STEM dataset (see Fig. RR2). The derived charge density map exhibits superior clarity compared to our ABF-STEM attempts. Although the reviewer questions the suitability of 4D-STEM for our objectives, we find the multi-faceted information it provides to be extremely valuable. This technique has indeed helped us address our scientific questions very effectively.

Fig. RR2: Four-dimensional STEM reconstructions of the interface structure between Fe and Fe₃O₄. **a** Dark-field (DF) image formed using a 101–124 mrad virtual aperture range. **b** Reverse bright-field (r-BF) image using a 20 mrad virtual aperture. **c** Reverse annular bright-field (r-ABF) image using a 10–20 mrad virtual aperture range. Each upper panel shows the reconstructed atomic-scale image, while the corresponding lower panels display the virtual apertures used to produce these images from the 4D-STEM

dataset. **d** A charge density map reconstructed from the same dataset, providing enhanced contrast of the atomic columns. The coordinate axes indicate the crystallographic orientations, and the Fe/FeO/Fe₃O₄ interface is highlighted.

Changes made in response to the comment:

Page 15, line 475, added: References 75-77

Page 15, line 478-480, added: ...as we find the multi-faceted information it provides to be valuable to help us address our scientific questions effectively.

Reviewer #3 Comment 3: Finally, in Fig. 4, the authors claim to observe a thick FeO-type interfacial phase but fail to show its precise thickness or details about the FeO/Fe₃O₄ interface. An interfacial phase is typically thin, as it forms only with the support of bulk phases. The reported thick condition in Fig. 4 is more characteristic of a bulk phase than an interfacial one.

Response: We appreciate the concern of the reviewer regarding the characterization and thickness of the FeO-type interfacial phase. The measured thickness, approximately 1 nm, is consistent with reported interfacial complexes, such as the ~1.2 nm films observed at gold–sapphire interfaces [M. Baram et al., Science 332, 206 (2011)]. While it is inherently challenging to determine exact thicknesses here, the observed value falls well within the typical range for interfacial complexes, rather than indicating a distinctly bulk-like phase.

To address the concerns from the reviewer, we have replaced the original Fig. 4 in the main text with new image coverage that more confidently demonstrates three- to four-atomic-layer-thick interfacial structures, presented now in Fig. RR3 (Fig.4 of the revised version of the paper). This adjustment aligns more closely with the known thickness ranges of such complexes and should address the concerns from the reviewer.

Fig. RR3: Different regions of the Fe/Fe₃O₄ interface displaying FeO-type complexes of various thicknesses. i DFT-computed electronic charge densities, **ii** DFT-relaxed equilibrium structure, and **iii** DPC-4DSTEM reconstructed charge density map for **a** three-layer and **b** four-layer FeO-type complexes at the Fe/Fe₃O₄ interface. The relaxed DFT structure of the interface is presented as a bridging sketch

between the two charge density maps in both **a** and **b**. (This figure corresponds to the updated Fig. 4 in the revised version of the manuscript.)

With respect to the original version of the manuscript, as also detailed in our response to **Reviewer #3 Comment 1** above, we now also considered new magnetic orders of the FeO-like complexions which differ from the original bulk AF order. For the three-layer and four-layer complexions, this yields a decrease of the interface energy by 37 and 8 meV/A², respectively. We further used the structures of the different interfaces corresponding to these new magnetic orders for comparison to experimental observations. The updated Fig. 4 is shown in Fig. RR3, as it appears in the revised version of the manuscript.

Changes made in response to the comment:

We have updated the original Fig. 4 to show the one with three-layer and four-layer complexions. Additional reconstruction information for the four-layer complexions has been added to the new Supplementary Fig. 15.

Page 10-11, line 329-332, added: Additional observations in Fig. 4, Supplementary Fig. 14 and Supplementary Fig. 15 corroborate this prediction, showing that in certain regions of the Fe/Fe₃O₄ interface, the three-layer and four-layer FeO-type complexion with are evident, respectively.

Reviewer #4: The authors have addressed most of the remarks that I had made except for the images in Figure R6. The images shown for the Fe₃O₄ oriented in the [210] zone axis are not in accord with the cubic phase for Fe₃O₄. I suggest the authors show the DPC images from Fe[110] and Fe₃O₄[100] orientations in the main text. Though the Fe overlaps with O in this orientations, the FeO-like phase can be distinguished from the Fe matrix by comparing the lattice parameter along [001]. The interfacial FeO-like phase can be confirmed clearly after the authors provide the variation of lattice parameter.

Response: We are grateful for the valuable comments and careful examination from the reviewer. The reviewer is absolutely correct that the Fe₃O₄ orientation was incorrectly noted. A typographical error led us to indicate the [210] zone axis, whereas the correct orientation is [310]. We appreciate this observation, which allowed us to rectify the inaccuracy.

Following the suggestions from the reviewer, we quantified the lattice parameters from the DPC image presented in Fig. RR4 below, considering viewing axis from Fe[110] and Fe₃O₄[100]. For the Fe phase, the FeO-type interfacial phase, and the Fe₃O₄ phase, we measure lattice parameters to be approximately 0.29 nm, 0.43 nm, and 0.83 nm, respectively (see black lines in Fig. RR4 below). These measurements are consistent with our DFT-predicted values, which are 0.28 nm, 0.43 nm, and 0.84 nm, respectively. To maintain focus in the main text, this updated figure will be provided in the Supplementary section. This approach ensures that readers who are particularly interested in the structural specifics have access to the information, while keeping the main text concise.

We trust that these changes address the concerns raised by the reviewer and improve the clarity and accuracy of our findings.

Additionally, as we measured the lattice constants on Fig. RR4, we realized the scale bars of the different experimental images were inaccurate. We corrected the scale bars accordingly in all figures presented in the revised version of this manuscript.

Fig. RR4. Experimental DPC-4DSTEM charge density reconstruction of the coherent interface between Fe and Fe₃O₄, with Fe₃O₄ oriented in the [100] direction. Approximate lattice parameters ($a_{\text{Fe}_3\text{O}_4}$, a_{FeO} and a_{Fe}) are highlighted as black lines between layers of atoms.

Changes made in response to the comment:

We have updated Supplementary Fig. 18 to include quantification of the lattice parameters of the Fe phase, the FeO-type interfacial phase, and the Fe₃O₄ phase.

Page 12, line 344-349, added: For the regions without misfit dislocations, we measured lattice parameters for the Fe phase, the FeO-type interfacial phase, and the Fe₃O₄ phase to be approximately 0.29 nm, 0.43 nm, and 0.83 nm, respectively (see black lines in Supplementary Fig. 18). These measurements are consistent with our DFT-predicted values of 0.28 nm, 0.43 nm, and 0.84 nm, respectively, confirming the existence of the interfacial FeO-type phase.